# Accounting for the climate-carbon feedback in emission metrics

Thomas Gasser[1,2,*], Glen P. Peters[2], Jan S. Fuglestvedt[2], William J. Collins[3], Drew T. Shindell[4], Philippe Ciais[1]

[1] Laboratoire des Sciences du Climat et de l'Environnement, LSCE/IPSL, Université Paris-Saclay, CEA – CNRS – UVSQ, F-91191 Gif-sur-Yvette, France
[2] Center for International Climate and Environmental Research – Oslo (CICERO), 0349 Oslo, Norway
[3] Department of Meteorology, University of Reading, Reading, RG6 6BB, UK
[4] Nicholas School of the Environment, Duke University, Durham, NC 27708, USA
* Now at: International Institute for Applied Systems Analysis (IIASA), A-2361 Laxenburg, Austria

*Correspondence to*: Thomas Gasser (gasser@iiasa.ac.at)

**Abstract.** Most emission metrics have previously been inconsistently estimated by including the climate-carbon feedback for the reference gas (i.e. $CO_2$) but not the other species (e.g. $CH_4$). In the fifth assessment report of the IPCC, a first attempt was made to consistently account for the climate-carbon feedback in emission metrics. This attempt was based on only one study, and therefore the IPCC ~~presented tentative values and~~ concluded that more research was needed. Here, we carry out this research. First, using the simple carbon-climate model OSCAR v2.2, we establish a new impulse response function for the climate-carbon feedback. Second, we use this impulse response function to provide new estimates for the two most common~~usual~~ metrics: Global Warming Potential (GWP) and Global Temperature change Potential (GTP). We find that, when the climate-carbon feedback is correctly accounted for, the emission metrics of non-$CO_2$ species increase, but in most cases not as much as initially indicated by IPCC. We also find that, when the feedback is removed for both the reference and studied species, the~~se~~ relative metric values only have modest changes~~,~~ compared to when the feedback is included (absolute metrics change more markedly). ~~However, including carbon-climate feedbacks, particularly in absolute metrics or for short time horizons, gives a more realistic representation of the response.~~ Including or excluding the climate-carbon feedback ultimately depends on the user's goal, but consistency should be ensured in either case.

## 1 Introduction

Emission metrics are a tool to compare or combine the climate impact of the emission of different greenhouse gases and other climate forcing agents, typically putting them on a so-called $CO_2$-equivalent scale. The physical meaning of this scale depends on the climate parameter chosen to calculate the metric (e.g. radiative forcing or temperature change), but also on the time-horizon and on whether it is an instantaneous or integrative metric. Emission metrics can be given in absolute terms or in relative terms, the latter being ~~a normalization of~~ the absolute metric ~~to~~taken relatively to that of a reference gas which is usually $CO_2$. For instance, GWP100 – the most widely used metric – is a ~~normalized~~ relative metric defined as the ratio of the cumulative radiative forcing induced after 100 years by 1 kg of a given species over that induced by 1 kg of $CO_2$. The GWP100

is currently used in UNFCCC emission inventories, climate agreements (e.g. the Kyoto Protocol), and climate policies (e.g. emissions trading systems). Emission metrics are also used to evaluate multi-gas policies, to compare emissions and sinks from countries and/or economic sectors, or simply as zeroth-order models of the climate system. They are used in areas such as life cycle assessment (e.g. Levasseur et al., 2016), ecosystem service study (e.g. Neubauer and Megonigal, 2015) and

integrated assessment modelling (e.g. Clarke et al., 2014). More about emission metrics can be found elsewhere (e.g. Cherubini et al.; 2016; Myhre et al., 2013; Shine et al., 2015).

Since emission metrics are based on simple representations of more complex models, there are choices in how components of complex models are incorporated in the metrics. One such component is the climate-carbon feedback. The "climate-carbon feedback" refers to the effect that a changing climate has onto the carbon-cycle, which impacts atmospheric $CO_2$, which in

turn changes further the climate. In concrete terms: when $CO_2$ is emitted, the atmospheric $CO_2$ pool increases. A fraction of this excess atmospheric $CO_2$ is taken up by the ocean and the terrestrial biosphere (the "carbon sinks"), but as long as a part of the excess $CO_2$ stays in the atmosphere, it warms the climate. In turn, this warming climate slows down the uptake of the atmospheric $CO_2$ by the sinks. This slowing down constitutes a positive feedback, i.e. a warming climate is warmed further through the feedback (Ciais et al., 2013). Rather than a slowing down of the carbon sinks, it is also possible to view the

feedback as a reduction of the carbon sinks uptake efficiency (Raupach et al., 2014). According to models of the coupled carbon-cycle – climate system, the climate-carbon feedback has contributed to the observed warming over the last century and will have a large impact in warmer future scenarios (e.g. Ciais et al., 2013; Friedlingstein et al., 2006; Raupach et al., 2014). Yet,although there are large uncertainties about the magnitude of this feedback and underlying mechanisms (e.g. Ciais et al., 2013; Friedlingstein et al., 2006; Raupach et al., 2014).

The standard metrics provided in the fifth assessment report (AR5) of the IPCC (Myhre et al., 2013; table 8.A.1) are inconsistent in their treatment of the climate-carbon feedback. While absolute metrics for $CO_2$ itself do account for the feedback, the absolute metrics for all other species do not. As a result, the normalized relative metrics, defined as the ratio of the absolute metric of a non-$CO_2$ species over that of $CO_2$, are inconsistently calculated. Aware of this limitation, the IPCC made a first attempt at including the climate-carbon feedback into metrics in a consistent manner. This attempt was based on

an earlier study by Collins et al. (2013) which whose main object was not the climate-carbon feedback (but regionalized metrics). This study is therefore an attempt to assess the robustness of these alternative but tentative metrics proposed by the IPCC (Myhre et al., 2013; table 8.7).

Here, we carry out an analysis of the climate-carbon feedback and how it can be included in the emission metrics framework. To do so, in section 2, we recall the mathematical framework used to derive emission metrics, and we extend it with a specific

term representing the response of the carbon sinks to climate change. In section 3, we use the simple Earth system model OSCAR v2.2 to derive a functional form for this response, and to quantify its numerical parameters. In section 4, we use the extended framework and our new response function to establish new values of metrics that include the climate-carbon feedback, and we compare those with the values otherwise available.

## 2 Mathematical framework

### 2.1 Impulse Response Functions

Emission metrics are usually formulated by means of Impulse Response Functions (IRFs), as it is done in the fifth IPCC report (Myhre et al., 2013). These IRFs are simple models which describe the dynamical response of a subsystem of the Earth system
(e.g. the biogeochemical cycle of a given species, or the climate system) to a pulse of perturbation of this subsystem. The response of the subsystem to a more general continuous and time-varying perturbation can be obtained by convolution of the IRF with the time-series of the perturbation. The various IRFs used are generally estimated on the basis of idealised simulations made with complex models (e.g. Geoffroy et al., 2013; Joos et al., 1996; 2013). Per construction, the IRFs are dynamical models which feature e.g. inertia and hysteresis, but they are linear in nature with respect to the intensity of the perturbation,
they represent a fully reversible system, and they can only include feedbacks in an implicit manner. Despite these apparent caveats, the use of such a linear-response approach to emulate the behaviour of complex systems can be warranted by the theory, especially in the case of the climate system (see e.g. Ragone et al., 2016; Lucarini et al., 2017). Note that emission metrics can also be estimated with more complex model simulations (e.g. Tanaka et al., 2009; Sterner and Johansson, 2017), with the strong caveat that the approach lacks the simplicity and transparency of the IRFs.
Now let us illustrate the typical formulation of the is simple IRF-based model in the case of the climate change induced by of a given species ($x$). The change in atmospheric concentration of the species ($Q^x$) can be calculated with a convolution between the time-series of anthropogenic emission of this species ($E^x$) and the IRF for the species' atmospheric concentration ($r_Q^x$):

$$Q^x(t) - Q^x(0) = \int_{t'=0}^{t} E^x(t')\, r_Q^x(t - t')\, dt'$$

In the most general case, the radiative forcing induced by this species ($RF^x$) is taken as a function ($\mathcal{F}^x$) of its change in
atmospheric concentration (e.g. Myhre et al., 1998):

$$RF^x(t) = \mathcal{F}^x\big(Q^x(t) - Q^x(0)\big)$$

And finally, the change in global mean surface temperature induced by this species ($T^x$) is again deduced by a convolution of the radiative forcing with the IRF for the climate system. This IRF is broken down into a dynamical term ($r_T$) and a constant intensity term ($\lambda$) that corresponds to the equilibrium climate sensitivity. This gives:

$$T^x(t) - T^x(0) = \lambda \int_{t'=0}^{t} RF^x(t')\, r_T(t - t')\, dt'$$

Typically, the IRF for atmospheric $CO_2$ is taken from Joos et al. (2013), those for other greenhouse gases are exponential decay functions with a constant e-folding time taken as the "perturbation lifetime" given by Myhre et al. (2013), the radiative forcing functions come from Ramaswamy et al. (2001) with updated radiative efficiencies from Myhre et al. (2013), and the climate

IRF is taken from Boucher and Reddy (2008). Note, however, that updates of the climate IRF based on CMIP5 models are available in the literature (Geoffroy et al., 2013; Olivié et al, 2013) but they have not been widely used so far.

## 2.2 Formulation of emission metrics

To produce emission metrics IRFs are used, albeit with two important additional assumptions. First, the initial anthropogenic perturbation is actually taken as a pulse of emission at time $t = 0$, which we can write formally with the Dirac-δ function and the size of the pulse ($E_0$) as follows: $E^x(t) = E_0^x \delta(t)$. Strictly speaking, the Dirac-δ is a distribution, and it is the (approximated) identity of the convolution algebra so that the convolution of any function by the Dirac-δ gives back the initial function. Second, since in the metrics framework this pulse is assumed to be very small, the radiative forcing function is approximated to be linear so that we have: $RF^x(t) = \varphi^x\big(Q^x(t) - Q^x(0)\big)\mathcal{F^x} = \varphi^x$, where $\varphi^x$ is the constant marginal radiative efficiency of the considered species. Note that the assumption of a very small pulse may be inconsistent with the way the IRFs are actually derived, as it is currently the case for $CO_2$ (see appendix A).

From there, we can formulate the Absolute Global Warming Potential (AGWP) and the Absolute Global Temperature-change Potential (AGTP), which are absolute (i.e. non-normalized) metrics. Per definition, the AGWP of a species $x$ is the cumulative radiative forcing induced by a pulse of emission of the species, normalized by the size of the pulse, and taken up to a chosen time horizon ($H$):

$$AGWP^x(H) = \frac{1}{E_0^x} \int_{t=0}^{H} RF^x(t)\, dt$$

$$= \frac{1}{E_0^x} \int_{t=0}^{H} \varphi^x \int_{t'=0}^{t} E_0^x\, \delta(t')\, r_Q^x(t - t')\, dt'\, dt$$

$$= \varphi^x \int_{t=0}^{H} r_Q^x(t)\, dt$$

Per definition, the AGTP of a species $x$ is the instantaneous temperature change induced by a pulse of emission of the species, normalized by the size of the pulse, and taken at a chosen time horizon:

$$AGTP^x(H) = \frac{1}{E_0^x} [T^x(H) - T^x(0)]$$

$$= \frac{1}{E_0^x} \lambda \int_{t=0}^{H} \varphi^x\, r_T(H - t) \int_{t'=0}^{t} E_0^x\, \delta(t')\, r_Q^x(t - t')\, dt'\, dt$$

$$= \varphi^x \lambda \int_{t=0}^{H} r_Q^x(t)\, r_T(H - t)\, dt$$

The Global Warming Potential (GWP) and the Global Temperature-change Potential (GTP) are metrics ~~normalized~~ calculated relatively to the reference gas $CO_2$. Therefore, any of these two metrics is defined as the ratio of its absolute counterpart for the species $x$ over that for $CO_2$:

$$GWP^x(H) = \frac{AGWP^x(H)}{AGWP^{CO_2}(H)}$$

and:

$$GTP^x(H) = \frac{AGTP^x(H)}{AGTP^{CO_2}(H)}$$

We can now detail the inconsistency mentioned in introduction, regarding the way the default GWPs and GTPs are estimated by the IPCC (Myhre et al., 2013; table 8.A.1). To estimate the absolute metrics for $CO_2$, the IRF derived by Joos et al. (2013) is used, and one feature of this IRF is that it implicitly includes any feedback between the climate system and the carbon-cycle
that is also included in the complex carbon-climate models it is calibrated upon. However, the absolute metrics for non-$CO_2$ species do not include the effect of the warming climate onto the carbon-cycle that is induced by the non-$CO_2$ species. In other words, the climate-carbon feedback is included in the denominator of the GWP and GTP, but *not* in their numerator. The resulting metric values should therefore be regarded as inconsistent.

## 2.3 Addition of the climate-carbon feedback

To include the climate-carbon feedback in the metric framework, we choose to model the decrease in the carbon sinks efficiency induced by climate change as an additional flux of carbon to the atmosphere, but without changing the atmospheric lifetime of carbon dioxide. Another approach, mathematically equivalent, would be to change the atmospheric lifetime of the gas. However, the latter approach cannot be used with the IRF framework since, per construction, the atmospheric lifetimes of all the species are fixed.

We define the change in the global carbon sinks $\Delta F$. It is positive if the flux goes into the atmosphere, i.e. if the sinks efficiency is actually reduced. By analogy with previous IRF-based equations, we propose the following formulation:

$$\Delta F^x(t) = \gamma \int_{t'=0}^{t} [T^x(t') - T(0)]\, r_F(t - t')\, dt'$$

In this equation, the forcing term is the global mean temperature change induced by the species $x$. The IRF for the carbon sinks is broken down into two terms: a dynamical term that is $r_F$, expressed in yr$^{-1}$; and an intensity term that is $\gamma$, expressed in GtC
yr$^{-1}$ K$^{-1}$. There are two implicit assumptions with this formulation which are discussed hereafter. First, we assume that the carbon sinks response is the same, at global scale and for a given temperature change, whatever the forcing species. Second, we assume that the global mean temperature is a proxy of all the changes in the climate variables that drive a change in the carbon sinks, such as local temperature itself but also precipitation.

To simplify the discussion and avoid quintuple integrals, we introduce the simplified notation $\star$ for the convolution: $a \star b \equiv \int_0^t a(t') \, b(t - t') \, dt'$, and note the commutative property of the convolution: $a \star b = b \star a$.

Since the change in carbon sinks is expressed as a new source of $CO_2$, one can calculate the additional radiative forcing ($\Delta RF$) induced by a species $x$ through the climate-carbon feedback:

$$\Delta RF^x = (\varphi^{CO2}) \, \Delta F^x \star r_Q^{CO2}$$

$$= (\varphi^{CO2} \, \gamma) \, [T^x - T(0)] \star r_F \star r_Q^{CO2}$$

$$= (\varphi^{CO2} \, \gamma \, \lambda) \, RF^x \star r_T \star r_F \star r_Q^{CO2}$$

$$= (\varphi^{CO2} \, \gamma \, \lambda \, \varphi^x) \, [Q^x - Q^x(0)] \star r_T \star r_F \star r_Q^{CO2}$$

$$= (\varphi^{CO2} \, \gamma \, \lambda \, \varphi^x) \, E^x \star r_Q^x \star r_T \star r_F \star r_Q^{CO2}$$

and similarly with the additional temperature change ($\Delta T$):

$$\Delta T^x = (\lambda) \, \Delta RF^x \star r_T$$

$$= (\varphi^{CO2} \, \gamma \, \lambda^2 \, \varphi^x) \, E^x \star r_Q^x \star r_T \star r_F \star r_Q^{CO2} \star r_T$$

We do not need to worry about the endless feedback loop $CO_2$–climate–$CO_2$ and add more terms to these equations, because the carbon dioxide IRF ($r_Q^{CO2}$) already accounts for the effect of climate change on the $CO_2$ concentration.

It is possible to formulate the additional absolute GTP ($\Delta AGTP$) – which is later added to the AGTP without feedback – for the species $x$:

$$\Delta AGTP^x = \frac{1}{E_0^x} \, \Delta T^x$$

$$= \left( \frac{1}{E_0^x} \, \varphi^{CO2} \, \gamma \, \lambda^2 \, \varphi^x \, E_0^x \right) \delta \star r_Q^x \star r_T \star r_F \star r_Q^{CO2} \star r_T$$

$$= (\gamma) \, r_F \star \underbrace{(\varphi^x \, \lambda) \, r_Q^x \star r_T}_{AGTP^x} \star \underbrace{(\varphi^{CO2} \, \lambda) \, r_Q^{CO2} \star r_T}_{AGTP^{CO2}}$$

that is:

$$\Delta AGTP^x(H) = \gamma \int_{t=0}^{H} r_F(H - t) \int_{t'=0}^{t} AGTP^x(t') \, AGTP^{CO2}(t - t') \, dt' \, dt$$

To formulate $\Delta AGWP$, it is easier to do the same demonstration if one introduces the Heaviside step function (i.e. the function equal to 1 for $t \geq 0$, and 0 otherwise; noted $\Theta$) and notes that convoluting any function with the Heaviside function is equivalent to integrating it. the definition of AGWP then is:

$$AGWP^x(H) = \frac{1}{E_0^x} \int_{t=0}^{H} RF^x(t) \, dt \equiv \frac{1}{E_0^x} \, RF^x \star \Theta$$

Hence, similarly to the case of $\Delta AGTP$, we have:

$$\Delta AGWP^x = \frac{1}{E_0^x} \, \Delta RF^x \star \Theta$$

$$= (\gamma) \, r_F \star \underbrace{(\varphi^x \, \lambda) \, r_Q^x \star r_T}_{AGTP^x} \star \underbrace{\varphi^{CO2} \, r_Q^{CO2} \star \Theta}_{AGWP^{CO2}}$$

that is:

$$\Delta AGWP^x(H) = \gamma \int_{t=0}^{H} r_F(H-t) \int_{t'=0}^{t} AGTP^x(t') \, AGWP^{CO2}(t-t') \, dt' \, dt$$

The above discussion holds in the case of species-dependent climate sensitivity parameters, i.e. if we have $\lambda^x$ instead of $\lambda$ to account for climate efficacies (e.g. Hansen et al., 2005). These two formulas, for $\Delta AGWP$ and $\Delta AGTP$, are similar to those given by Collins et al. (2013) in their section 5.5, where they implicitly assume that: $\gamma \, r_F(t) = \Gamma \, \delta(t)$, where $\Gamma$ is a constant. Collins et al. (2013) therefore assumes that the carbon sinks response to a pulse of global temperature change was a pulse of size $\Gamma$ of $CO_2$ outgassing by the ocean and the terrestrial biosphere, but did not justify this assumption. The next section investigates whether this assumption holds, and what functional form can be chosen for the dynamical function $r_F$.

## 3 Estimating the climate-carbon feedback response

### 3.1 Experimental setup

We use the compact Earth system model OSCAR v2.2 to establish the IRF of the climate-carbon feedback (Gasser et al., 2016). It embeds several modules dedicated to simulating the response of many subsystems of the Earth system; and more specifically to our case, it embeds modules for the oceanic carbon-cycle, the terrestrial carbon-cycle and the climate system. Each of these modules is designed to emulate the sensitivity of more complex – usually spatially explicit – models. In the version used here, the complex models used to calibrate OSCAR were used for the IPCC AR5 via the Coupled Model Intercomparison Project phase 5 (CMIP5). OSCAR includes the following climate-carbon feedbacks: the effect of temperature and precipitation change on net primary productivity of land ecosystems, their heterotrophic respiration, and the rate of occurrence of wildfires; and the effect of temperature change on the carbonate chemistry and the stratification of the surface ocean. OSCAR is used in a probabilistic setup, which means that ensembles of simulations are made so as to be able to derive an uncertainty distribution for our results. These Monte Carlo ensembles contain 1200 elements; each element being the outputs of a simulation done with a set of parameters drawn with equiprobability from the pool of available parameterizations of OSCAR (Gasser et al., 2016). The configuration used here is similar to the one called "offline" by Gasser et al. (2016), and more information as to the basic performance of the model is also provided therein.

Before estimating the IRF for the climate-carbon feedback, we benchmark OSCAR's IRFs of the carbon-cycle and climate system separately against commonly used IRFs. For the carbon-cycle, we follow the protocol by Joos et al. (2013), reproduced in appendix A, and we repeat it a second time while turning off all the climate-carbon feedbacks of the model. The two carbon dioxide IRFs obtained are shown in figure 1a. The IRF obtained when the feedbacks are turned on is very close to the one derived by Joos et al. (2013) and used by the IPCC. When the feedbacks are turned off, the IRF decays faster than when they

are on, which means that the carbon sinks are more efficient – as expected. Regarding the climate response, since OSCAR's climate module is a two-box model with constant coefficients, it is equivalent to an IRF, shown in figure 1b. The model's response is close to the average of sixteen CMIP5 models as calculated by Geoffroy et al. (2013), but it differs from the one used in the IPCC AR5 (Boucher and Reddy, 2008). Together the ability of the OSCAR model to reproduce the carbon-cycle and climate IRFs derived from up-to-date and complex models suggests that it is also capable of establishing a reasonable IRF for the climate-carbon feedback.

To estimate this climate-carbon feedback IRF, we adopt a protocol largely inspired by that of Joos et al. (2013) for the carbon dioxide IRF. A first simulation is made to calculate the background conditions, in which atmospheric $CO_2$ and non-$CO_2$ radiative forcings are prescribed up to 2010 exactly as it is done with the first simulation of the protocol for the carbon dioxide IRF (see appendix A). These prescribed forcings are then maintained for another 1000 years of simulation. The climate variables simulated in this first experiment are saved to be used in the second simulation. In OSCAR, these variables are the air surface temperature (global and regional over land), the sea surface temperature (global), and precipitation (global and regional over land). A second simulation is made in which the same atmospheric $CO_2$ and non-$CO_2$ radiative forcings are prescribed, along with the climate variables saved previously. In this second experiment, in the year 2015 and afterwards, a constant climate perturbation is added on top of the prescribed climate from the first experiment. This perturbation has a global average surface temperature change of +0.2°C, but the local temperature and precipitation perturbations do vary spatially, following the response patterns used in OSCAR and calibrated on complex models (Gasser et al., 2016). In our model, these regional response patterns are easy to obtain, since they are proportional to the global average temperature change, but for more complex models the protocol might have to be adapted (see discussion). Finally, the climate-carbon feedback response (not yet the IRF of section 2.3) is calculated as the difference between the global $CO_2$ flux from the oceanic and terrestrial carbon reservoirs to the atmosphere simulated in the second and first experiments, normalized by the size of the global temperature step, and setting the time origin ($t = 0$) as the starting year of the step (i.e. 2015).

## 3.2 Results

Figure 2 shows the carbon sinks' response to the temperature step change simulated by OSCAR v2.2. Panel (b) shows the model change in surface flux due to decreased carbon sinks, panel (a) shows the cumulative response from summing the flux and panel (c) shows the differentiated response from taking the year-to-year difference in flux. If the yearly response is the "speed" of outgassing of the carbon sinks, the differentiated response is its "acceleration". It is important to note that the analytical time-step of OSCAR is one year, and that it is not a process-based model. It is thus impossible to specifically distinguish the very short-term response of the carbon sinks to the step of climate change. Despite this limitation, over the period of time we can study, the response simulated by OSCAR is very different from that assumed by Collins et al. (2013). In OSCAR, the response of the carbon sinks to a step of climate change is an instantaneous burst of outgassing followed by more outgassing that is however decreasing in intensity with time, despite the constant intensity of the forcing (figure 2b). We also find the land carbon flux response is about double that from the ocean (not shown). This response is physically very

different from Collins et al. (2013) and thus the IPCC, where it is assumed that the carbon sinks response to a pulse of climate change is a pulse of outgassing, or equivalently that their response to a step of climate change is a step of outgassing. This would imply that under a stabilized but changed climate (e.g. at +2°C on global average) the carbon sinks would endlessly release $CO_2$ to the atmosphere. This is unrealistic, since the total emitted $CO_2$ is limited by the size of the natural reservoirs.

Our simulations show the carbon sinks behaving in a more reasonable and expected way. Under a step of climate change, the sinks do release $CO_2$ – which is consistent with the positive sign of the climate-carbon feedback – but the release of $CO_2$ slows down with time (figure 2b), until the sinks reach a new equilibrium under a new climate. This behaviour implies that the total amount of released $CO_2$ is capped (figure 2a) and is given by the difference in the natural carbon pools between the two equilibria under the two different climatic backgrounds. The response to a pulse of climate change is indeed a burst of

outgassing; however, after the pulse, the atmospheric $CO_2$ is now raised above the equilibrium level so the sinks increase, eventually recapturing the lost carbon (figure 2c). The latter part of the response was missing from Collins et al. (2013).

### 3.3 Estimating the IRF

In this section, we estimate a functional form for the climate-carbon feedback IRF that will then be used to estimate new emission metrics. We look only at the time period covered by our simulations with OSCAR, therefore ignoring the

discontinuity around $t = 0$. Let us call $f$ the function of the time variable that will fit the simulated cumulative response (figure 2a). The yearly response (figure 2b) is thus fitted by $f'$ – its first derivative – and the differentiated response (figure 2c) by $f''$ – its second derivative. The functional form of $f$ is chosen to be a sum of three saturating exponential functions, consequently:

$$f(t) = \gamma \left( \alpha_1 \tau_1 \left( 1 - \exp\left(-\frac{t}{\tau_1}\right) \right) + \alpha_2 \tau_2 \left( 1 - \exp\left(-\frac{t}{\tau_2}\right) \right) + \alpha_3 \tau_3 \left( 1 - \exp\left(-\frac{t}{\tau_3}\right) \right) \right)$$

$$f'(t) = \gamma \left( \alpha_1 \exp\left(-\frac{t}{\tau_1}\right) + \alpha_2 \exp\left(-\frac{t}{\tau_2}\right) + \alpha_3 \exp\left(-\frac{t}{\tau_3}\right) \right)$$

$$f''(t) = -\gamma \left( \frac{\alpha_1}{\tau_1} \exp\left(-\frac{t}{\tau_1}\right) + \frac{\alpha_2}{\tau_2} \exp\left(-\frac{t}{\tau_2}\right) + \frac{\alpha_3}{\tau_3} \exp\left(-\frac{t}{\tau_3}\right) \right)$$

Each of the three exponentials is parameterized by a time constant $\tau_i$ and a weight $\alpha_i$, and the overall function is also parameterized by its intensity $\gamma$. The $\gamma$ parameter is introduced here by choice, and it is the same as in section 2.3. Since we introduce a seventh parameter while only six were needed (we could have defined three $\gamma_i$ as $\gamma_i = \gamma \, \alpha_i$), we also add the constraint that $\alpha_1 + \alpha_2 + \alpha_3 = 1$. The choice of an exponential-based functional form is motivated by the fact that all other IRFs typically

used for emission metrics are also formulated with exponentials, because it allows closed-form analytical solutions of all the convolutions. Another interest of exponential-based IRFs is the possibility to use Laplace transforms to study the carbon-climate system (Enting, 2007).

To deduce numerical values for the parameters, we fit the $f$ function and its first and second derivatives over the three response curves simulated by OSCAR and shown in figure 2. ~~To do so, we use only the actual outputs of OSCAR, i.e. the fit is made~~

~~only over the simulated curves and not over the extended ones.~~ To determine the six freely-varying parameters, we proceed in four steps that are detailed in appendix B. Table 1 shows the parameters obtained by repeating the procedure for the average, upper and lower responses of the ensemble. The intensity parameter of the response ($\gamma$) is ~3.0 GtC $yr^{-1}$ $K^{-1}$. The three time constants of the carbon sinks response are consistent with the atmospheric $CO_2$ response of OSCAR, but there is more weight placed on the faster modes so that the carbon response to a temperature pulse is faster than the carbon response to a $CO_2$ pulse. However, it is extremely difficult to relate any of the physical processes to these parameters (Li et al., 2009). We also tried other functional forms for this fit, specifically forms with fewer exponentials, but it was not possible to capture both the dynamics of the first few years and that of the last hundreds of years.

The response obtained with OSCAR exhibits a discontinuity around $t = 0$ (figure 2) as the model cannot simulate the response of the carbon sinks over short time-scales (<1 yr). We assume nonetheless that the flux perturbation can be extrapolated back to $t = 0^{\pm}$, neglecting any processes faster than a year that we cannot represent. Thus the discontinuity at $t = 0$ is modelled with a Dirac-$\delta$ function whose intensity is equal to the value of the flux at $t = 0^+$. The resulting extension of the simulated response is schematically shown in figure 2. ~~and~~

Based on the above, we can finally propose a mathematical expression of the climate-carbon feedback IRF defined in section 2.3:

$$\gamma \, r_F(t) = \ f'(0^+)\delta(t) + f''(t)$$

$$= \gamma\delta(t) - \gamma\left(\frac{\alpha_1}{\tau_1} \, \exp\left(-\frac{t}{\tau_1}\right) + \frac{\alpha_2}{\tau_2} \, \exp\left(-\frac{t}{\tau_2}\right) + \frac{\alpha_3}{\tau_3} \, \exp\left(-\frac{t}{\tau_3}\right)\right)$$

The constraint $\alpha_1 + \alpha_2 + \alpha_3 = 1$ implies that $\int_{0^-}^{+\infty} r_F(t) = 0$. This means that, in our framework, a pulse of climate change has no effect on the natural carbon pools on the very long-term. In other words, in the response shown in figure 2c, the (infinite) recovery period fully compensates for the initial pulse of $CO_2$ emission. This idealised feature of reversibility is to be expected from the simple and linear modelling framework that the impulse response functions are, since no multiple equilibria is permitted. This is however likely unrealistic, given all the existing ~~non-linear~~ processes, such as vegetation migration (e.g. Jones et al., 2009) or permafrost thawing (e.g. Koven et al., 2011), that can produce some degree of irreversibility in the system but are ignored here.

**3.4 Influence of step size and background conditions**

To assess the robustness of our IRF, as well as its domain of validity, we repeat the simulations with different steps of temperature. We derive IRFs for climate change steps corresponding to a global mean temperature increase of: +0.01, +0.1, +0.2, +0.5, +1, +2, +3, +4, +5 and +10 °C. We note however that for the latter values, and especially for +10°C, we are pushing the model into a domain where its performance is questionable. The parameters we obtain for each experiment are shown in figure 3. The climate-carbon feedback intensity ($\gamma$) decreases when the step size increases. Since the intensity is normalized by the step size, this does not mean the feedback is weaker when climate change is stronger. This rather means the carbon

sinks response is non-linear in intensity: a doubled step of climate change induces less than a doubled outgassing of the natural reservoirs. This saturation effect can be explained by the limited size of the reservoirs: the fewer carbon remains, the harder it is to get it out (i.e. the more energy is required). The climate-carbon feedback effective time-scale ($\tau_{\text{eff}}$; calculated as $\tau_{\text{eff}} = \sum_i \alpha_i \tau_i$) also decreases when the step size increases, indicating that under a stronger climate change perturbation the carbon sinks outgassing occurs faster. These two non-linear behaviours appear small for the very small perturbations (i.e. below +1°C). We also repeat the simulations with different background conditions, though only for climate change steps corresponding to a global mean temperature increase of +0.2 and +1 °C. Four different background conditions are obtained with a slight alteration of our protocol: the background-setting part of the simulation – i.e. before the step of climate change – is extended to follow each Representative Concentration Pathway (RCP) atmospheric $CO_2$ and radiative forcing data (Meinshausen et al., 2011) from 2005 to 2100, and the step occurs in 2105 instead of 2015. Figure 3 shows that the higher the atmospheric $CO_2$ and global warming of the background, the more intense and faster the climate-carbon feedback, with a doubling of the intensity parameter ($\gamma$) and a decrease by one-third of the time-scale parameter ($\tau_{\text{eff}}$) under RCP8.5. These results can be explained by the increased amount of carbon stored in the natural reservoirs at the time of the climate change step, as in the model the carbon sinks keep removing $CO_2$ from the atmosphere during the RCP simulation while atmospheric $CO_2$ is higher than today. These results are also consistent with those regarding the atmospheric $CO_2$ IRF (Joos et al., 2013): under a higher $CO_2$ and temperature background, it is harder for the carbon sinks to remove $CO_2$ from the atmosphere (slower carbon dioxide IRF) and it is easier for them to release the carbon they are already storing (stronger and faster feedback IRF). Both studies – that of Joos et al. (2013) and ours – therefore show that the carbon-cycle is a non-linear system that can be only approximatively emulated by impulse response functions.

**4 New estimates of emission metrics**

Using the estimated IRF for the climate-carbon feedback, we now provide new estimates of the two most common emission metrics, GWP and GTP, for five species spanning a broad range of atmospheric lifetimes and climate impacts: methane ($CH_4$), nitrous oxide ($N_2O$), sulphur hexafluoride ($SF_6$), black carbon (BC) and sulphur dioxide ($SO_2$). We follow the methodology used by the IPCC in the AR5 (Myhre et al., 2013): we use the perturbation lifetimes for non-$CO_2$ species and the radiative efficiencies they provide (their table 8.A.1), the carbon dioxide IRF from Joos et al. (2013), and the climate IRF from Boucher and Reddy (2008). For BC and $SO_2$, because the IPCC does not provide a unique set of parameters for these short-lived species, we choose the globally averaged ones from Fuglestvedt et al. (2010). We also have to settle on one of our climate-carbon feedback IRFs: we choose the one corresponding to present-day background conditions and a global climate change step of +0.2°C. This choice is motivated by the fact that +0.2°C is approximatively the globally averaged peak warming induced by a pulse of $CO_2$ emission of 100 GtC which is itself the value chosen by Joos et al. (2013)— and ~~therefore by~~ used in the IPCC AR5. We then use the equations given in section 2.3, solving the convolutions numerically with a time-step of one tenth of a

year. Figure 4̶5̶ is provided as an illustration of this process whereby we calculate the ΔAGTP of methane, starting from the initial pulse of $CH_4$ and going through the five successive convolutions described earlier.

The metrics values are shown in figures 5̶6̶ (AGWPs and AGTPs) and 6̶7̶ (GWPs and GTPs). In these figures, we show separately the default IPCC metrics (Myhre et al., 2013; table 8.A.1) and the additional effect of the climate-carbon feedback (i.e. the Δ-term that will then be added to the metrics) obtained with both the Collins et al. (2013) formulation and ours. The Δ-terms always act to increase the magnitude of both the absolute and relative climate metrics. Although the Δ-terms from Collins et al. (2013) are of similar orders of magnitude, their function forms are very different. Since Collins et al. (2013) did not include the re-uptake of carbon following the initial pulse, their Δ-terms keep increasing with the time horizon, while ours peak and decline. Eventually, the Collins et al. Δ-term is even larger than the default metric on long timescales, which is never the case with our formulation. Note that there is no Δ-term for $CO_2$ as the climate-carbon feedback is already included in the default metrics, hence including it in the metrics for non-$CO_2$ species restores consistency.

In table 2 (first three rows) we show the climate metrics, including and excluding Δ-term, for three chosen time horizons: 20, 50 and 100 years. There, one can see again that the metrics are systematically higher (in absolute value) than in the default IPCC case, when the climate-carbon feedback induced by non-$CO_2$ species is accounted for, whatever the chosen formulation. Quantitatively, however, for long time horizons, the IPCC (Myhre et al., 2013; table 8.7), based on Collins et al. (2013), overestimates the effect of the climate-carbon feedback, whereas this effect is underestimated for short time horizons. This can also be seen in figures 5̶6̶ and 6̶7̶ where the dotted lines are below the dashed ones during the first decades, and over afterwards. In table 2 (fourth row), we also provide new estimates of the metrics including the climate-carbon feedback as calculated with OSCAR, but also a̶n̶d̶ with the climate IRF updated from that of Boucher and Reddy (2008) to that of Geoffroy et al. (2013). The latter is calibrated on several climate models of the latest generation, while the former appears to be an outlier of the CMIP5 ensemble – see our figure 1b and results for "HadGEM2-ES" provided by Geoffroy et al. (2013). In concrete terms, the IRF of Boucher and Reddy (2008), used by the IPCC, is slower but has a higher climate sensitivity than the one calibrated on the CMIP5 multi-model mean. The effect of this update can be seen by comparing the third and fourth rows of our table 2. Updating the climate IRF has more effect on the GTPs than on the GWPs, which is logically due to the fact that GTP is defined as a function of the temperature (see section 2.2) while GWP is a function of the radiative forcing and is therefore affected by the temperature only through the climate-carbon feedback. Changing the climate IRF impacts the GTPs for all species, but for short-lived species (BC and $SO_2$, and to a lesser extent $CH_4$) a revised climate IRF has an effect as large as correcting the climate-carbon feedback term. This is a reminder of the sensitivity of the GTPs to the representation of the climate time-scales (in $r_T$), and that these are at least as important as including or neglecting the climate-carbon feedback.

In table 2 (fifth row), we provide another set of relative metrics, similar to the previous one in that it includes the feedback response calibrated on OSCAR and the updated climate IRF, but it also includes an update of the radiative efficiencies of $CO_2$, $CH_4$ and $N_2O$ (Etminan et al., 2016). The new radiative efficiency of $CO_2$ differs by +2%, that of $CH_4$ by +14%, and that of $N_2O$ by -3%. These changes logically impact the GWPs and the GTPs, since both metrics are function of the $\varphi^x$ parameters. The change is substantial for $CH_4$: in most cases more so than the update of the climate IRF. Notably, the update of the radiative

efficiency of $CO_2$ – being the reference gas in relative metrics – implies a change in the metrics' values of all species, even those whose own radiative efficiency are not changed. These results show that the first-order processes (here, the radiative forcing) may have more impact on the metrics than second-order processes such as the climate-carbon feedback.

~~We recommend using the metrics shown in this fourth row of table 2, since they are the most consistent, robust and up to date metrics available. Analytical expressions of the IRFs, to be used to calculate metrics for other time horizons and/or other species, are given in appendix C.~~

In table 2 (last two rows), to fully understand the effect of including or not the climate-carbon feedback in emission metrics, we provide two other sets of metrics: the two are based only on IRFs derived from OSCAR (i.e. the responses shown in figures

1 and 2), with one including the feedback for both $CO_2$ and non-$CO_2$ while the other does not for either. In both cases (i.e. when the climate-carbon feedback is consistently included or excluded) the metrics are very close. For greenhouse gases (here: $CH_4$, $N_2O$ and $SF_6$) the difference remains below 10%, with only very small changes for the GWPs. Only in the case of the GTP of short-lived species (BC and $SO_2$) and for short time horizons is the difference larger than that, reaching about 30%.

Finally, we show in table 3 that the relative uncertainties associated with these OSCAR-based metrics – calculated using our

Monte Carlo ensembles and uncertainty ranges from Myhre et al. (2013; table 8.SM.12) – remain close, no matter whether the climate-carbon feedback is included or not, as long as it is consistent. This can be explained by the fact that the climate-carbon feedback only makes a small contribution to the climate metrics. Therefore, despite being highly uncertain, it does not contribute much to the overall uncertainty.

**5 Discussion and conclusion**

We have developed a theoretical framework to consistently include the climate-carbon feedback in emission metrics, we have used the simple model OSCAR v2.2 to establish an IRF for the feedback, and finally, we have used the framework and the new IRF to propose new estimates of the GWP and GTP. The overarching goal of our study was to correct and complement the work initiated by Collins et al. (2013) and reflected by the IPCC, and to provide a framework that could be used in future IPCC assessment reports. To this end, we see two technical points that must be discussed: one regarding the underlying

assumptions made when we extend the IRF framework to include the climate-carbon feedback; and one regarding the possibility of applying our protocol to more complex models. And to conclude, we open up the discussion to more general considerations about the IRF framework and the interest (or lack thereof) of accounting for the climate-carbon feedback in emission metrics, and about the role of non-$CO_2$ species in the global climate system.

**5.1 Technical aspects**

In our extended metrics framework, to account for the climate-carbon feedback, we link the global mean temperature change to the global total change in carbon removal by the natural sinks. This global approach averages over differing regional

responses. Consequently, the causal links between i) global climate change and local climate changes, and ii) local climate changes and local responses of the ocean or land sinks are accounted for only implicitly with our modelling approach. Regarding the first causal link, since we apply the same IRF ($\gamma\, r_F$) whatever the forcing species $x$, we implicitly assume that the local pattern of climate change is always the same whatever. This is certainly not the case in reality for temperature (e.g.

Hansen et al., 2005) or precipitation (e.g. Shine et al., 2015); note thatand the latter affects the land sink. This could be addressed by repeating our experiment with different patterns of temperature and precipitation corresponding to various forcers so as to deduce species-dependent IRFs in the form, for instance, of a set of $\gamma^x$ parameters. Regionally varying climate responses have been explored by e.g. Shindell and Faluvegi (2009) and Collins et al. (2013) and could in principle be used to generate species dependent $r_F$, although they are very uncertain. Regarding the second causal link, i.e. from local climate change to

local carbon sinks response, the local response to climate change can be of a sign different from the global one, and further altered if nutrients such as nitrogen are accounted for (Ciais et al., 2013). Therefore, if IRFs were established at the regional scale, they would not likely resemble the one shown in figure 2.

We have established an analytical expression for the climate-carbon feedback IRF with a simple carbon-climate model and following a specific protocol. Although OSCAR performs well in simulating historical changes in the global Earth system

(Gasser et al., 2016) and in calculating carbon dioxide and climate IRFs (see figure 1), our simulations should be reproduced with other – more complex – carbon-climate models, to check whether our results hold qualitatively and quantitatively. Ideally, a multi-model modelling exercise such as the one that led to the carbon dioxide IRF (Joos et al., 2013) should also include the simulations required to establish the climate-carbon feedback IRF. For a complex carbon-cycle model, the step climate change could be defined as the difference between the end of a CMIP $4\times CO_2$ experiment and the control experiment (simulated by

the same model). Note that step changes rather than gradual changes such as +1%/yr $CO_2$ increase (e.g. Arora et al., 2013) are needed in order to derive the IRFs.

## 5.2 Conceptual aspects

In a more general perspective, our results raise the question of whether the climate-carbon feedback should be included in emission metrics. Accounting for the feedback implies more simulations in a multi-model exercise similar to that of Joos et al.

(2013) for calibration purposes, whereas not accounting for it requires a new set of $CO_2$ IRFs with the feedback turned off. For most greenhouse gases (e.g. CH$_4$, N$_2$O, SF$_6$), we found that inclusion of the climate-carbon feedback does not greatly change (less than 10%) the values of the normalized GWPs and GTPs provided the feedback is included consistently for CO$_2$ and non-CO$_2$ species. For very short-lived species (e.g. SO$_2$, BC) the feedback does have a significant effect over short time-horizons (greater than 30%). The absolute metrics do change substantially when including climate-carbon feedback. Including the

climate-carbon feedback therefore gives consistency and accuracy across a wide range of species and time-horizons. We have found that including or excluding the climate-carbon feedback in a consistent manner does not greatly change the values of the relative GWPs (only about 2%). In the case of relative GTPs, the change is slightly larger for greenhouse gases (less than 10%) and becomes even larger for very short-lived species and over short time-horizons (greater than 30%). In the case of

absolute metrics – both AGWPs and AGTPs – these changes are substantial since we are adding a positive feedback to the model. Therefore, the choice of including or excluding the feedback ultimately depends on the user's needs. On the one hand, for the sake of simplicity and transparency, the feedback could be excluded from the evaluation of GWPs, since it avoids the trouble of the five convolutions shown in figure 4. On the other hand, if absolute (e.g. time-varying) metrics are used as a first-order model of climate change, one may prefer including the climate-carbon feedback to have a better representation of the system. We provide in appendices C and D all the analytical expressions needed to calculate the metrics with or without the feedback.

It is also important to note that the above changes in the metrics' value are of the same order of magnitude (and sometimes less) as the change induced by the update of the climate IRF and the radiative efficiencies of greenhouse gases, as shown in section 4. Hence multiple types of physical properties need to be correctly accounted for. They are also less in magnitude than those induced by the choice of the protocol used to calculate the metrics, such as the background conditions (e.g. Reisinger et al., 2011), or by the choice of a given time horizon (see e.g. table 2). Although these factors reflect choices about temporal applicability of the metrics rather than refined understanding of physical behaviour.

If the choice is made that this feedback be included in emission metrics, it then raises another question as to what other feedbacks, if any, should also be included. Let us take the climate-wetlands feedback as an example. When climate changes, so does the amount of $CH_4$ emitted by natural wetlands (e.g. Ciais et al., 2013). This could be included in a manner similar to what we did with the climate-carbon feedback: the atmospheric $CH_4$ IRF should be re-calculated with interactive wetlands, and a new IRF for the climate-wetlands feedback induced by non-$CH_4$ forcers should be established. This is feasible; but now one must consider that wetlands emissions are also directly affected by atmospheric $CO_2$ through $CO_2$-fertilisation and altered stomatal closure that alters the local hydrological cycle (Ciais et al., 2013). Therefore, accounting for the carbon-climate-wetlands nexus requires a much more complex experimental setup. And this is just one example: feedbacks involving biogeochemical cycles in the Earth system are numerous (Ciais et al., 2013). It can be rightfully argued that some of these feedbacks can be neglected, and that others can be safely linearized (such as the $CH_4$-OH feedback that is included in emission metrics in the AR5). Nevertheless, it appears that we are reaching the limits of the IRF framework which is linear by essence. The alternative, to include all the possible feedbacks in emission metrics, is actually to develop model-based estimates similarly to what is done for atmospheric chemistry, for instance to look at species-species interactions (e.g. Shindell et al., 2009), regional specificities (e.g. Collins et al., 2013) or the seasonality of processes and drivers (e.g. Aamaas et al., 2016). However, this ~~would be~~is at the expense of the simplicity and transparency that are characteristic of the impulse response functions. For the climate-carbon feedback, Sterner and Johansson (2017) recently proposed a first model-based estimate. Their results show the same difference in physical behaviour when compared to Collins et al. (2013) as ours, therefore strengthening our conclusions as to the need to update the IPCC metrics' estimates.

It could also be argued that, rather than concentrating on improving the level of detail in representing the typical climate impacts associated with GWP and GTP (i.e. radiative forcing and global temperature change, respectively), it would be more

useful if metrics were instead expanded to more comprehensively capture the full range of environmental impacts associated with emissions, such as extreme events, crop yields or air pollution (e.g. Shindell, 2015).

~~But u~~Ultimately, the new IRF we established also sheds some light on the climate-carbon feedback and on the role of non-$CO_2$ species in the global climate system. Using a simple model, a robust framework and idealised experiments, we complement earlier studies on the climate-carbon feedback (e.g. Friedlingstein et al., 2006; Arora et al., 2013) with new qualitative insights as to the dynamics of the feedback. This complex dynamics – summed up in our figure 4~~5~~ – has the peculiar effect of giving a long-term impact to short-lived species. Therefore, our work shows that non-$CO_2$ species have an additional impact on the global climate system through this feedback loop, as others showed before (e.g. Gillet and Matthews, 2010; Mahowald, 2011; MacDougall and Knutti, 2016). It must be understood, however, that this "enhancement" of the non-$CO_2$ species' impact – as called by MacDougall and Knutti (2016) – does not actually imply that non-$CO_2$ species are comparatively more important, in the context of climate change mitigation, than initially though. In fact, while it is true that the climate impact of non-$CO_2$ species is increased via the climate-carbon feedback (i.e.~~–~~ their absolute metrics are increased)~~—~~ so is the climate impact of $CO_2$ alone; so that the relative importance of non-$CO_2$ species *versus* $CO_2$ when the feedback is included for both remains surprisingly close to the case in which the feedback is not included (i.e.~~–~~ their ~~normalized~~ relative metrics remain similar).

## 5.3 Concluding remarks

As pointed out in the IPCC AR5, the metric calculations should consistently include the same processes for both $CO_2$ (denominator) and non-$CO_2$ emissions (numerator). We have explored including the climate-carbon feedback in both, and revised the preliminary calculations presented in the AR5. Given the complexities of the climate-carbon feedback, it would be beneficial to have more studies, with models of varying complexity, to verify our conclusions. Given that inclusion of the climate-carbon feedback has the greatest impact on metrics with short-lived climate forcers, it would be especially interesting to examine the impact of their inhomogeneous distributions on the spatial pattern of the climate-carbon response.

~~To avoid potential biases in metric values, we suggest to include the climate-carbon feedback in metric estimates.~~ Ultimately, whether emission metrics should include the climate-carbon feedback is a decision for the user, and we only recommend consistency in the way feedbacks are included or excluded. The trade-off between simplicity and transparency on the one hand, and accuracy of representation on the other hand, has to be weighed by the final user. But metric users ~~must~~ should also keep in mind that IRFs and emission metrics are extremely simple models of a complex system, and that sometimes it may be beneficial to use more complex models that better capture multiple and interacting feedback processes.

## Appendix A: Protocol to simulate the carbon dioxide IRF

The protocol is exactly that of Joos et al. (2013), reproduced here for clarity.

A first simulation is made in a concentration-driven fashion, with prescribed atmospheric $CO_2$ and prescribed non-$CO_2$ radiative forcings that follow the estimates by Meinshausen et al. (2011) for the historical period up to 2005, and then those

for the RCP4.5 between 2005 and 2010. These prescribed forcings are then maintained constant to their value of the year 2010 during another 1000 years of simulation. In the case of OSCAR, as recommended by Joos et al. (2013), land-use and land-cover change is also prescribed following the historical and then RCP4.5 data of Hurtt et al. (2011), and then stopped after 2010. The outputs from this first simulation are used to deduce the anthropogenic emissions of $CO_2$ that are compatible with

the prescribed atmospheric $CO_2$, through simple mass balance of the carbon element (see e.g. Gasser et al., 2015).

A second simulation is made in an emission-driven fashion with the same prescribed non-$CO_2$ radiative forcings and with the compatible $CO_2$ emissions deduced from the first simulation, with the only purpose of checking that the atmospheric $CO_2$ concentration simulated is the same as the one prescribed in the first simulation.

A third and final simulation is made, similar to the second one except that in 2015, on top of the compatible emissions, a pulse

of 100 giga-tonnes of carbon is added to the atmosphere. The carbon dioxide IRF seen in figure 1a is simply deduced as the difference between the atmospheric $CO_2$ simulated in the third and second experiments, normalized by the size of the pulse, and setting the time origin ($t = 0$) as the year of the pulse (i.e. 2015).

Specific to our study, we also make simulations following this protocol but with the climate-carbon feedbacks "turned off". This is achieved by prescribing the climate simulated by the second experiment to the third one.

**Appendix B: Protocol to fit the climate-carbon feedback IRF**

First, we fit a first-guess value for $\tau_1$, using the differentiated response (figure 2c) only over the first 5 (annual) time-steps, and assuming that $f''$ can be approximated by a one-exponential function over this short period of time. Second, we fit a first-guess value for $\gamma$ and $\alpha_1$, using the yearly response (figure 2b) also over the first 5 time-steps, and assuming that $f'$ can also be approximated by a one-exponential function whose time constant $\tau_1$ is the one estimated during the first step. Third, we fit a

first-guess value for the remaining parameters, i.e. $\tau_2$, $\tau_3$ and $\alpha_2$, using the cumulative response (figure 2a) over the whole simulation, and using the parameters determined in the first and second steps for $f$. Fourth, we fit the final values of the six parameters, using the yearly response (figure 2b) but this time over the whole simulation, and using the six parameters previously estimated as first guesses of the parameters of $f'$. All fits follow a least squares method, with the additional constraints that: $0 < \alpha_i < 1$ and $\alpha_3 = 1 - \alpha_1 - \alpha_2$. Only the actual outputs of OSCAR are used to fit, i.e. the 'extended' part shown

in figure 2 is not used.

**Appendix C: Analytical expressions of the IRFs used in this study**

**C.1 Carbon dioxide response**

Joos et al. (2013):

$$r_Q^{CO_2}(t) = 0.2173 + 0.2763 \exp\left(-\frac{t}{4.304}\right) + 0.2824 \exp\left(-\frac{t}{36.54}\right) + 0.2240 \exp\left(-\frac{t}{394.4}\right)$$

OSCAR v2.2, with climate-carbon feedback (average of ensemble):

$$r_Q^{CO_2}(t) = 0.2366 + 0.2673 \exp\left(-\frac{t}{4.272}\right) + 0.2712 \exp\left(-\frac{t}{33.10}\right) + 0.2249 \exp\left(-\frac{t}{302.4}\right)$$

OSCAR v2.2, without climate-carbon feedback (average of ensemble):

$$r_Q^{CO_2}(t) = 0.2033 + 0.3016 \exp\left(-\frac{t}{4.736}\right) + 0.2836 \exp\left(-\frac{t}{34.09}\right) + 0.2115 \exp\left(-\frac{t}{288.4}\right)$$

## C.2 Climate response

Boucher and Reddy (2008):

$$\lambda\, r_T(t) = 1.06 \left(\frac{0.595}{8.4} \exp\left(-\frac{t}{8.4}\right) + \frac{0.405}{409.5} \exp\left(-\frac{t}{409.5}\right)\right)$$

Geoffroy et al. (2013):

$$\lambda\, r_T(t) = 0.885 \left(\frac{0.587}{4.1} \exp\left(-\frac{t}{4.1}\right) + \frac{0.413}{249} \exp\left(-\frac{t}{249}\right)\right)$$

OSCAR v2.2 (average of ensemble):

$$\lambda\, r_T(t) = 0.852 \left(\frac{0.572}{3.50} \exp\left(-\frac{t}{3.50}\right) + \frac{0.428}{166} \exp\left(-\frac{t}{166}\right)\right)$$

## C.3 Climate-carbon feedback response

Collins et al. (2013):

$$\gamma\, r_F(t) = 1.0\, \delta(t)$$

OSCAR v2.2 (average of ensemble):

$$\gamma\, r_F(t) = 3.015\, \delta(t) - 3.015 \left(\frac{0.6368}{2.376} \exp\left(-\frac{t}{2.376}\right) + \frac{0.3322}{30.14} \exp\left(-\frac{t}{30.14}\right) + \frac{0.0310}{490.1} \exp\left(-\frac{t}{490.1}\right)\right)$$

## Appendix D: Other parameters used in this study

### D.1 Radiative efficiencies

The following values include the effect of any overlap between the absorption bands of $CO_2$, $CH_4$ and $N_2O$ (Myhre et al., 1998; Etminan et al., 2016). They also include some indirect effects: increase in stratospheric water vapor and tropospheric ozone for $CH_4$, and enhancement of the methane atmospheric sinks for $N_2O$ (Myhre et al., 2013; sections 8.SM.11.3.2 and 8.SM.11.3.3). Note that these indirect effects are not affected by the update of the direct radiative efficiency by Etminan et al. (2016). The background concentration is kept to that of 2011, as in IPCC AR5 (Myhre et al., 2013; section 8.SM.11.1). Myhre et al. (2013):

$$\varphi^{CO2} = 1.76 \times 10^{-15} \text{ W m}^{-2} \text{ kgCO2}^{-1}$$

$$\varphi^{CH4} = 2.11 \times 10^{-13} \text{ W m}^{-2} \text{ kgCH4}^{-1}$$

$$\varphi^{N2O} = 3.57 \times 10^{-13} \text{ W m}^{-2} \text{ kgN2O}^{-1}$$

$$\varphi^{SF6} = 2.20 \times 10^{-11} \text{ W m}^{-2} \text{ kgSF6}^{-1}$$

Etminan et al. (2016):

$$\varphi^{CO2} = 1.79 \times 10^{-15} \text{ W m}^{-2} \text{ kgCO2}^{-1}$$

$$\varphi^{CH4} = 2.39 \times 10^{-13} \text{ W m}^{-2} \text{ kgCH4}^{-1}$$

$$\varphi^{N2O} = 3.46 \times 10^{-13} \text{ W m}^{-2} \text{ kgN2O}^{-1}$$

Fuglestvedt et al. (2010):

$$\varphi^{SO2} = -3.2 \times 10^{-10} \text{ W m}^{-2} \text{ kgSO2}^{-1}$$

$$\varphi^{BC} = 1.96 \times 10^{-9} \text{ W m}^{-2} \text{ kg}^{-1}$$

**D.2 Perturbation lifetimes**

These are used to define the non-$CO_2$ atmospheric concentration IRFs: $r_Q^x(t) = \exp(-t/\tau^x)$.

Myhre et al. (2013):

$$\tau^{CH4} = 12.4 \text{ yr}$$

$$\tau^{N2O} = 121 \text{ yr}$$

$$\tau^{SF6} = 3200 \text{ yr}$$

Fuglestvedt et al. (2010):

$$\tau^{SO2} = 0.011 \text{ yr}$$

$$\tau^{BC} = 0.020 \text{ yr}$$

**Acknowledgements**

This work was partially funded by a visiting researcher grant from the Research Council of Norway (#249972). T.G. was also supported by the European Research Council Synergy project IMBALANCE-P (grant ERC-2013-SyG-610028). G.P.P., W.J.C., D.T.S. and J.S.F. were supported by the Research Council of Norway (project #235548).

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

| | $\gamma$ (GtC yr$^{-1}$ K$^{-1}$) | $\tau_{eff}$ (yr) | $\alpha_1$ (–) | $\alpha_2$ (–) | $\alpha_3$ (–) | $\tau_1$ (yr) | $\tau_2$ (yr) | $\tau_3$ (yr) |
|---|---|---|---|---|---|---|---|---|
| **mean** | **3.015** | **28.69** | **0.6368** | **0.3322** | **0.0310** | **2.376** | **30.14** | **490.1** |
| **upper** | 4.264 | 32.06 | 0.5916 | 0.3679 | 0.0405 | 3.333 | 37.12 | 404.3 |
| **lower** | 1.964 | 27.28 | 0.8139 | 0.1761 | 0.0100 [§] | 1.181 | 38.02 | 1962. [§] |

**Table 1: Values of the parameters of the IRF for the climate-carbon feedback (i.e. $\gamma\, r_F$).** The parameters calibrated on OSCAR for the mean response are given, as well as those for the upper response (i.e. mean +1 standard deviation) and the lower response (i.e. mean –1 standard deviation). The latter two responses correspond to the two envelope curves in figure 2. The effective time-scale $\tau_{eff}$ is calculated as $\tau_{eff} = \sum_i \alpha_i \tau_i$. [§] The low weight and high value of the slow time-scale indicate that the lower response could be fitted by a two-exponential functional form.

**(part 1)**

| Time horizon (in years) | CH₄[†] GWP 20 | 50 | 100 | GTP 20 | 50 | 100 | N₂O GWP 20 | 50 | 100 | GTP 20 | 50 | 100 | SF₆ GWP 20 | 50 | 100 |
|---|---|---|---|---|---|---|---|---|---|---|---|---|---|---|---|
| **AR5 (default)** [§] | 84 | 48 | 28 | 67 | 14 | 4 | *263* | 275 | *264* | *276* | *281* | 234 | 17500 | 20500 | *23600* |
| **AR5 + Collins** [§] | *85* | *52* | 34 | 70 | 20 | 11 | *267* | 290 | *297* | *283* | 311 | 297 | 17800 | 21600 | 26200 |
| **AR5 + OSCAR** | 86 | 52 | 31 | 70 | 18 | 5 | 269 | 289 | 283 | 285 | 304 | 258 | 17900 | 21500 | 25200 |
| **AR5 + OSCAR + climate IRF update** | 86 | 51 | 31 | 60 | 14 | 7 | 270 | 288 | 281 | 294 | 300 | 253 | 18000 | 21500 | 25000 |
| **AR5 + OSCAR + IRF & REs updates** | 96 | 57 | 34 | 67 | 16 | 7 | 256 | 274 | 267 | 279 | 285 | 240 | 17600 | 21100 | 24500 |
| **All OSCAR** | 96 | 57 | 34 | 66 | 18 | 9 | 255 | 273 | 267 | 279 | 283 | 241 | 17600 | 21000 | 24500 |
| **All OSCAR (no CC-fdbk)** | 96 | 57 | 34 | 65 | 16 | 8 | 257 | 275 | 269 | 282 | 286 | 244 | 17700 | 21200 | 24800 |

**(part 2)**

| Time horizon (in years) | SF₆ GTP 20 | 50 | 100 | BC[‡] GWP 20 | 50 | 100 | GTP 20 | 50 | 100 | SO₂[‡] GWP 20 | 50 | 100 | GTP 20 | 50 | 100 |
|---|---|---|---|---|---|---|---|---|---|---|---|---|---|---|---|
| **AR5 (default)** [§] | *19000* | *23900* | *28300* | 1560 | 736 | 426 | 451 | 71 | 58 | -140 | -66 | -38 | -40 | -6 | -5 |
| **AR5 + Collins** [§] | 19400 | 26000 | 33700 | 1620 | 818 | 519 | 528 | 172 | 165 | -145 | -73 | -47 | -47 | -15 | -15 |
| **AR5 + OSCAR** | 19500 | 25500 | 30800 | 1630 | 794 | 465 | 525 | 110 | 69 | -146 | -71 | -42 | -47 | -10 | -6 |
| **AR5 + OSCAR + climate IRF update** | 20500 | 25900 | 30400 | 1630 | 787 | 460 | 210 | 116 | 90 | -146 | -71 | -41 | -19 | -10 | -8 |
| **AR5 + OSCAR + IRF & REs updates** | 20100 | 25400 | 29800 | 1600 | 772 | 451 | 206 | 114 | 88 | -143 | -69 | -41 | -18 | -10 | -8 |
| **All OSCAR** | 20100 | 25200 | 29400 | 1590 | 769 | 450 | 213 | 147 | 105 | -143 | -69 | -40 | -19 | -13 | -9 |
| **All OSCAR (no CC-fdbk)** | 20400 | 25600 | 30200 | 1570 | 760 | 448 | 165 | 128 | 101 | -141 | -68 | -40 | -15 | -11 | -9 |

**Table 2: GWPs and GTPs at a time horizon of 20, 50 and 100 years, in the case of CH₄, N₂O, SF₆, BC and SO₂.** The first row ("AR5 default") shows the base metrics as calculated by the IPCC AR5 (Myhre et al., 2013; table 8.A.1). The second row ("AR5 + Collins") shows the metrics proposed in the IPCC AR5 as a first attempt to account for the climate-carbon feedback (their table 8.7), in which case the climate-carbon feedback IRF ($\gamma\, r_F$) is the one of Collins et al. (2013). The third row ("AR5 + OSCAR") shows the metrics when using our climate-carbon feedback IRF. The fourth row ("AR5 ~~updated~~ + OSCAR + climate IRF update") shows the same metrics as the third row, except that the climate IRF ($\lambda\, r_T$) is updated ~~from that of Boucher and Reddy (2008)~~ to one based on an ensemble of CMIP5 models~~that of~~ (Geoffroy et al.~~,~~ (2013). ~~This fourth row, in bold font, shows our recommended values.~~ The fifth row ("AR5 + OSCAR + IRF & REs updates") is the same as the fourth one, except that we also update the radiative efficiencies (REs) of CO₂, CH₄ and N₂O (Etminan et al., 2016). The sixth~~fifth~~ row ("all OSCAR") shows the metrics obtained when all IRFs used are based on OSCAR and the radiative efficiencies are also updated, with inclusion of the climate-carbon feedback. The seventh~~sixth~~ and last row ("all OSCAR no CC-fdbk") shows the same as the sixth~~fifth~~ row, but this time without including the climate-carbon feedback: neither for CO₂ nor for non-CO₂ species. [§] Because we use a numerical resolution method while the IPCC used an analytical one, some values in these rows may differ from the IPCC values by 1 because of the rounding (by 100 in the case of SF₆); these differing values are shown in italic font. [†] This does not account for the oxidation of CH₄ into CO₂ (see e.g. Boucher et al., 2009). [‡] Metrics for BC and SO₂ are not directly provided by the IPCC, rather we use here the IPCC method with lifetimes and radiative efficiencies from Fuglestvedt et al. (2010).

| | CH$_4$[†] | | | | | | N$_2$O | | | | | |
|---|---|---|---|---|---|---|---|---|---|---|---|---|
| | GWP | | | GTP | | | GWP | | | GTP | | |
| Time horizon (in years) | 20 | 50 | 100 | 20 | 50 | 100 | 20 | 50 | 100 | 20 | 50 | 100 |
| **All OSCAR** | 17% | 20% | 22% | 23% | 32% | 32% | 10% | 13% | 15% | 14% | 17% | 19% |
| **All OSCAR (no CC-fdbk)** | 19% | 22% | 24% | 24% | 34% | 34% | 13% | 16% | 18% | 15% | 19% | 21% |

**Table 3: Uncertainty of GWP and GTP at a time horizon of 20, 50 and 100 years, in the case of CH$_4$ and N$_2$O.** The relative uncertainties for ±1 standard deviation are shown. They are calculated on the basis of: i) the Monte Carlo ensembles of simulations made with OSCAR, shown in figures 1 and 2 and described in main text, and ii) the uncertainty ranges given by Myhre et al. (2013; table 8.SM.12) for radiative efficiencies and perturbations lifetimes. [†] This does not account for the oxidation of CH$_4$ into CO$_2$.

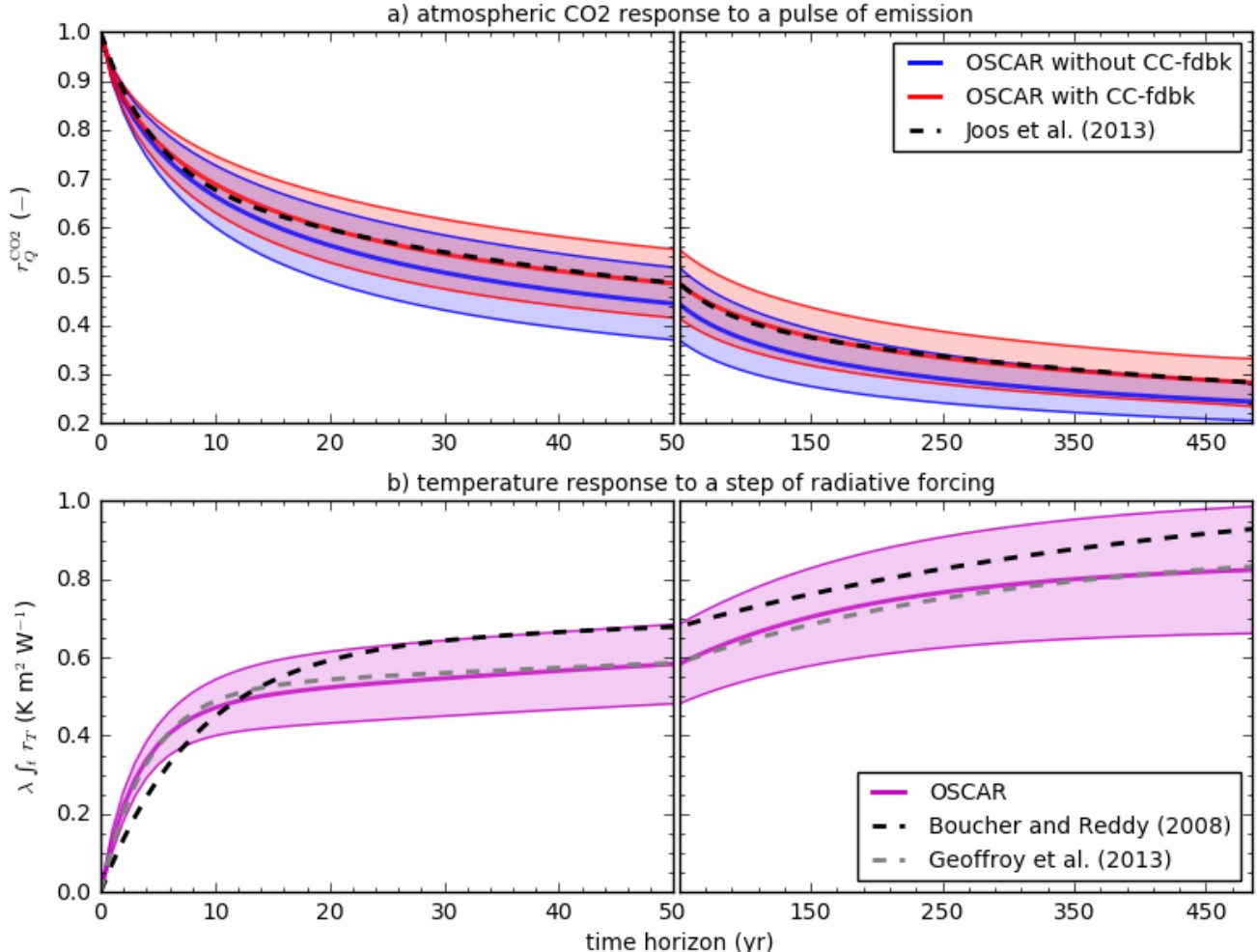

**Figure 1: Impulse response functions estimated with OSCAR.** (a) Response of the atmospheric $CO_2$ to a pulse of emission, in the case where the climate-carbon feedbacks ("CC-fdbk") are turned off (in blue), and in the normal case (in red). The responses by OSCAR are compared to that of Joos et al. (2013) used by the IPCC AR5 (dashed black). (b) Response of the global mean surface temperature to a step of radiative forcing. The response by OSCAR is compared to that of Boucher and Reddy (2008) used by the IPCC AR5 (dashed black) and to that of Geoffroy et al. (2013) that is based on CMIP5 models (dashed grey). The actual climate IRF (i.e. the response to a pulse) is obtained by taking the derivative of the curve shown in (b). Plain and thick lines show the mean response of OSCAR, while shaded and coloured areas show the ±1 standard deviation around the mean.

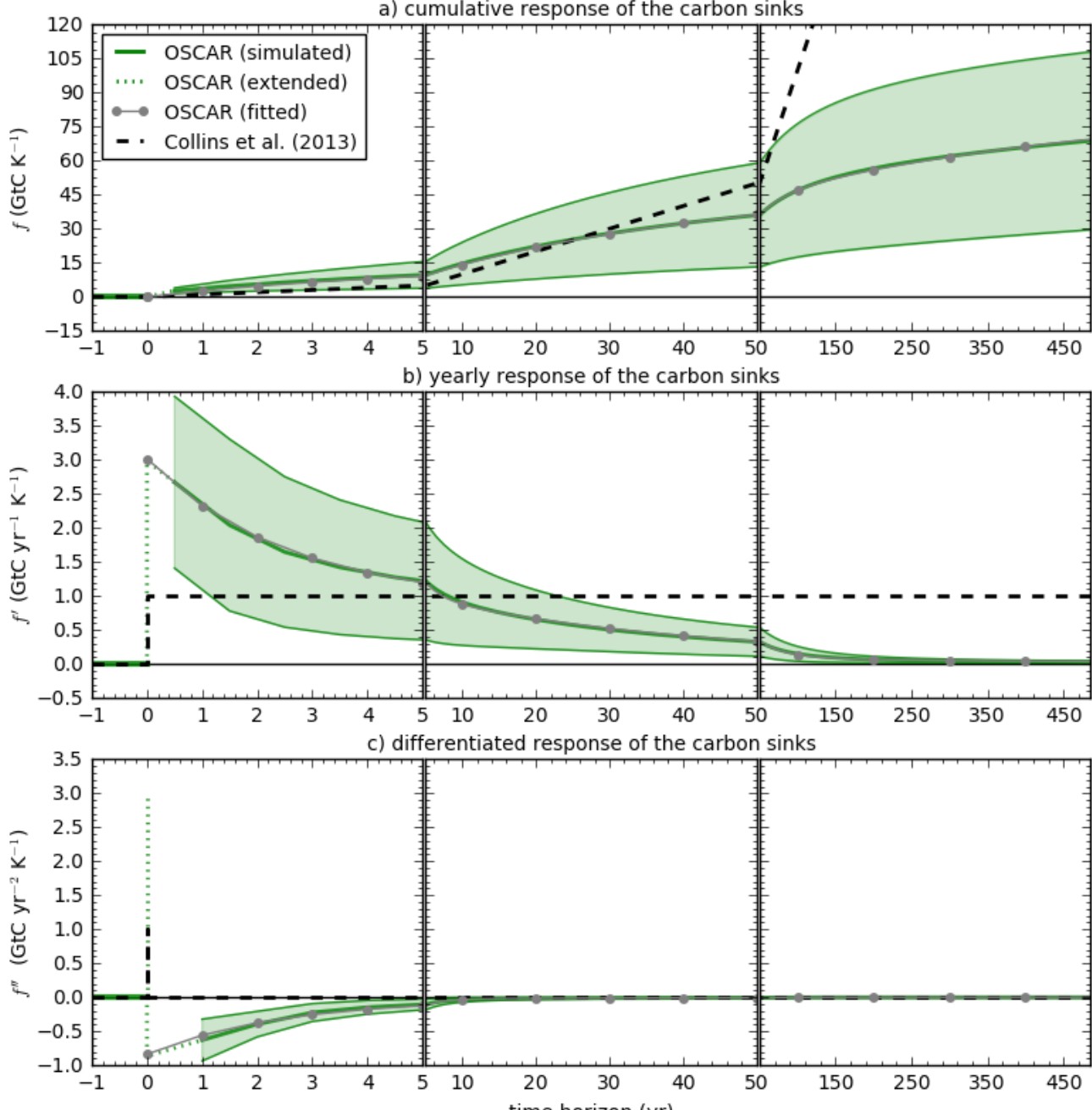

**Figure 2: IRF for the carbon sinks response estimated with OSCAR.** The response of the carbon sinks to a step of climate change is shown in three different ways: (a) as the cumulative amount of $CO_2$ outgassed by the sinks; (b) as the annual amount of $CO_2$ outgassed by the sinks; (c) as the derivative of the annual response to the step of climate change, which is equivalent to the annual response to a pulse of climate change. As in figure 1, the plain and thick (green) lines show the mean response from the Monte Carlo ensemble, while the shaded areas show the ±1 standard deviation. The dotted (green) lines illustrate our arbitrary extension of the response simulated by OSCAR when around $t = 0$ (see section 3.3). The grey lines with round markers are the results of our fit. For comparison, we also show the response assumed by Collins et al. (2013) as dashed black lines.

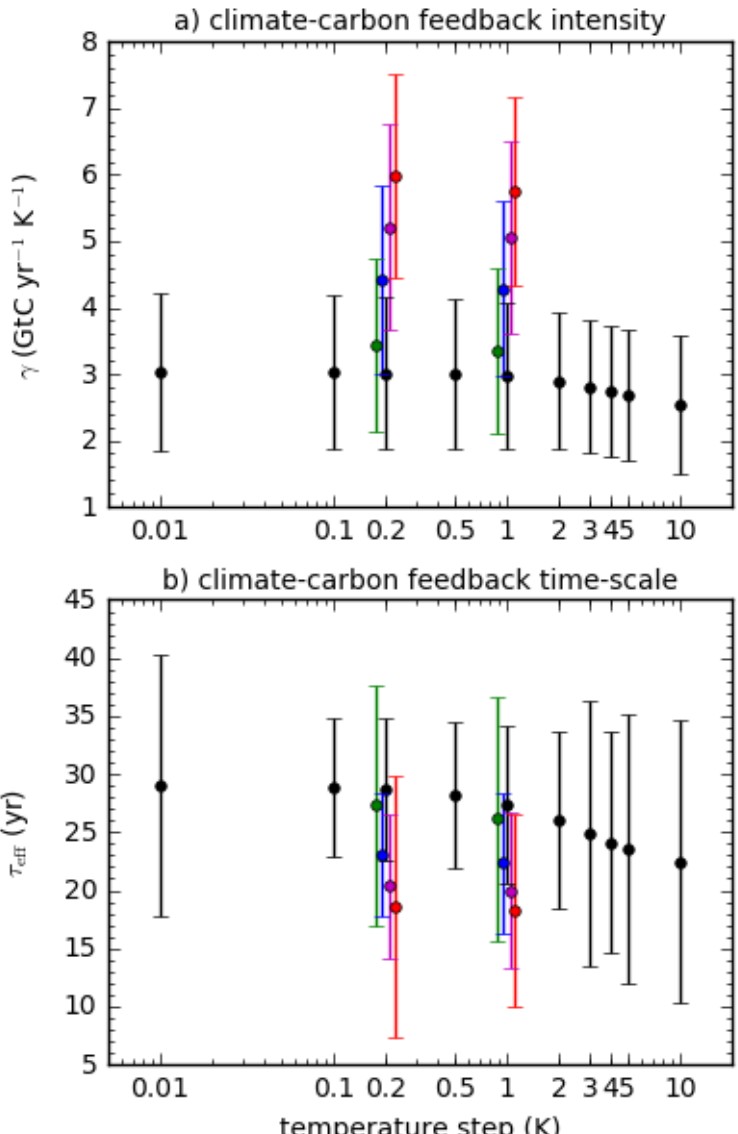

**Figure 3: Influence of step size and background on the climate-carbon feedback IRF:** (a) on the climate-carbon feedback intensity γ; and (b) on the climate-carbon feedback effective time-scale $\tau_{\text{eff}}$ (calculated as $\boldsymbol{\tau_{\text{eff}} = \sum_i \alpha_i \tau_i}$). The effect of the amplitude of the step of climate change (in black) and of the atmospheric $CO_2$ and climate background following the four RCPs (in colour; green for RCP2.6, blue for RCP4.5, magenta for RCP6.0 and red for RCP8.5) are shown. The uncertainty ranges shown is the ±1 standard deviation range, corresponding to the "upper" and "lower" responses in table 1.

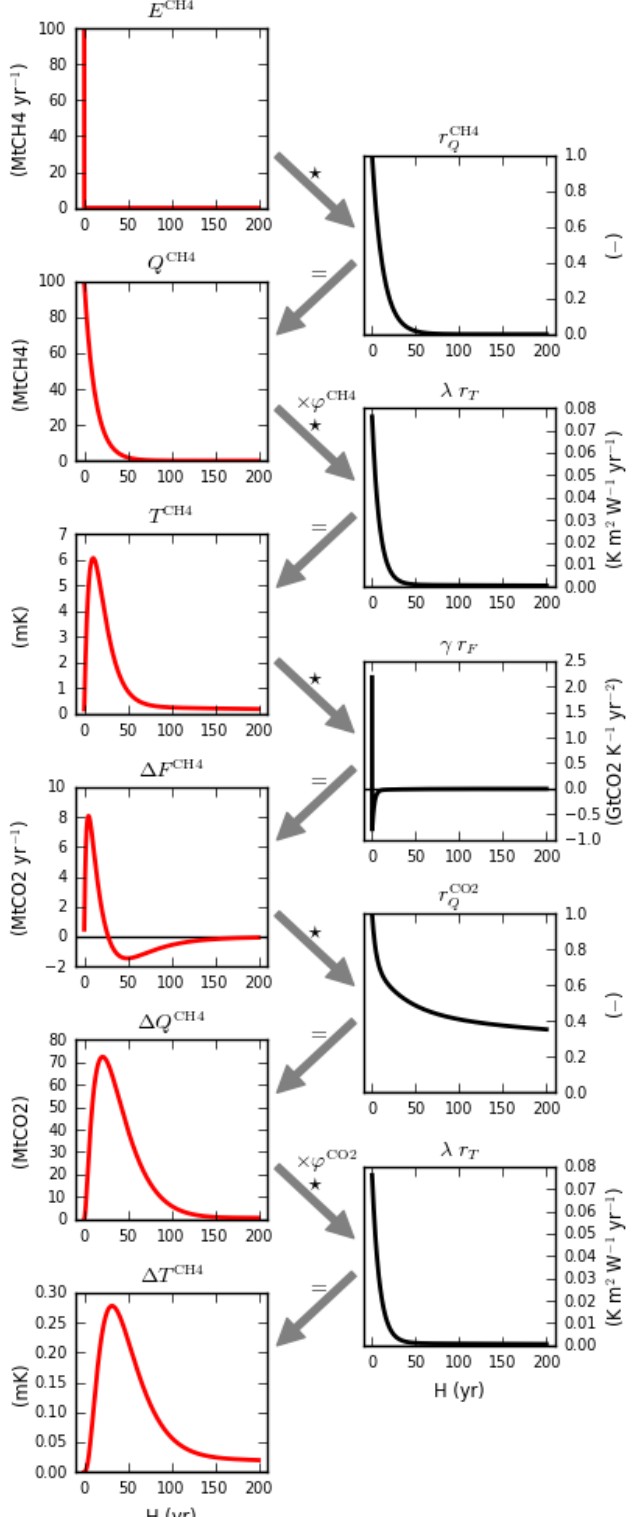

**Figure 4: Example of the step-by-step convolutions leading to the ΔAGTP of CH4.** The figure is read panel by panel, following the arrows and starting in the upper-left corner. The left-hand side panels show the actual physical variables, whereas the right-hand side panels show the IRFs used for the convolutions. We start with a pulse of CH4 emitted at $t = 0$, of an arbitrary size of 100 MtCH4. This pulse ($E^{CH4}$) is then convoluted (symbol $\star$) with the atmospheric CH4 IRF ($r_Q^{CH4}$) to give the induced change in atmospheric CH4 ($Q^{CH4}$). This atmospheric CH4 is then multiplied by the CH4 radiative efficiency ($\varphi^{CH4}$~~; units: W m⁻²GtCH4⁻¹~~) and convoluted with the climate IRF ($\lambda\, r_T$) to give the induced change in global surface temperature ($T^{CH4}$). One would stop here to deduce the AGTP by normalizing the obtained temperature change by the size of the initial pulse. In our case, the temperature change is then convoluted with the climate-carbon feedback IRF ($\gamma\, r_F$) to give the induced flux of $CO_2$ released by the sinks ($\Delta F^{CH4}$). This flux of $CO_2$ is then convoluted with the carbon dioxide IRF ($r_Q^{CO2}$) to give the induced change in atmospheric $CO_2$ ($\Delta Q^{CH4}$). And finally, this atmospheric $CO_2$ is then multiplied by the $CO_2$ radiative efficiency ($\varphi^{CO2}$~~; units: W m⁻² GtCO2⁻¹~~) and convoluted with the climate IRF ($\lambda\, r_T$) to give the induced change in global surface temperature ($\Delta T^{CH4}$). The ΔAGTP is deduced by normalizing the obtained temperature change by the size of the initial pulse. An analogous example can be produced for ΔAGWP, in which case one has to replace the last convolution by a convolution with the Heaviside step function (Θ).

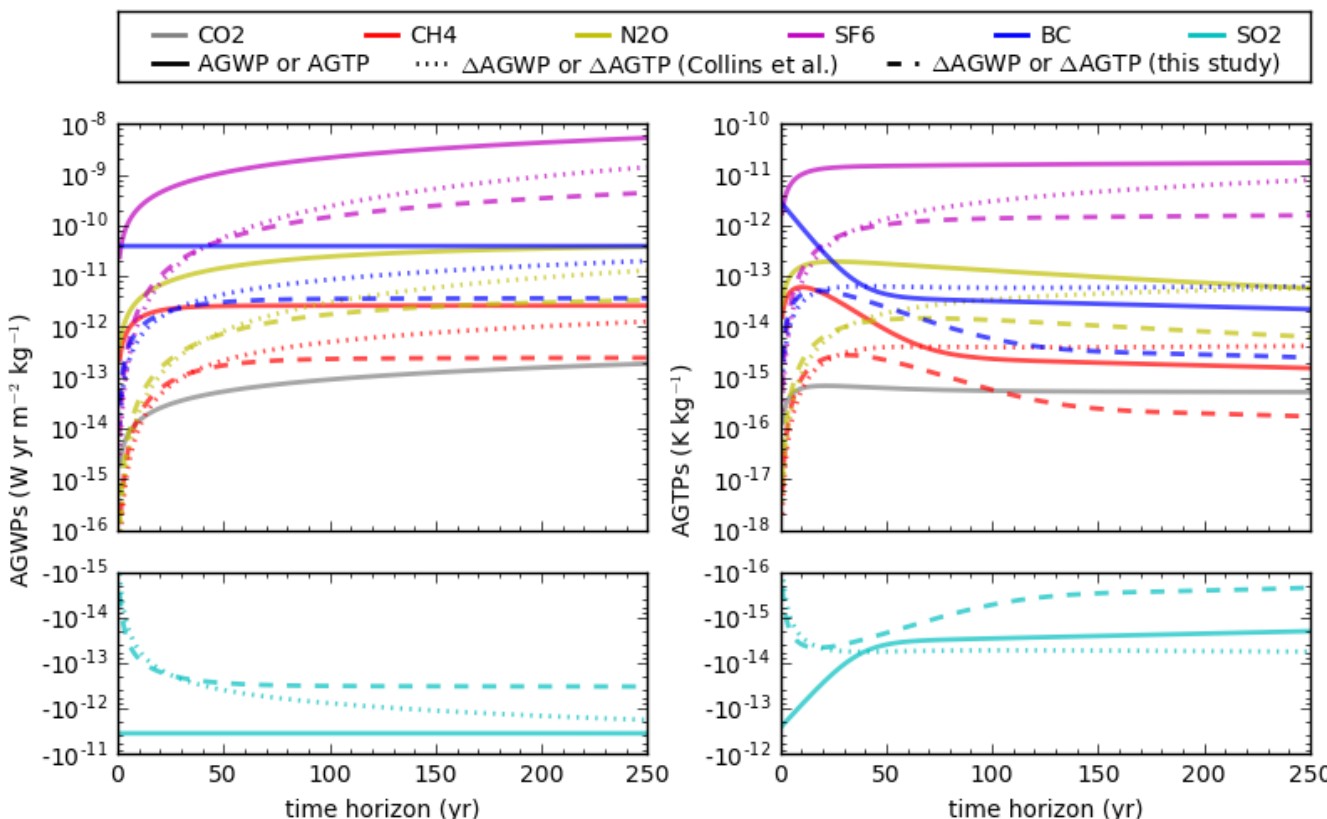

**Figure 5: Absolute metrics, in the case of CO₂, CH₄, N₂O, SF₆, BC and SO₂.** AGWPs (left-hand side) and AGTPs (right-hand side) obtained using the IPCC AR5 method are shown in solid lines. ΔAGWPs and ΔAGTPs obtained using the climate-carbon feedback IRF by Collins et al. (2013) are shown in dotted lines, and those obtained using ours are in dashed lines. Colours refer to the different species taken here as examples.

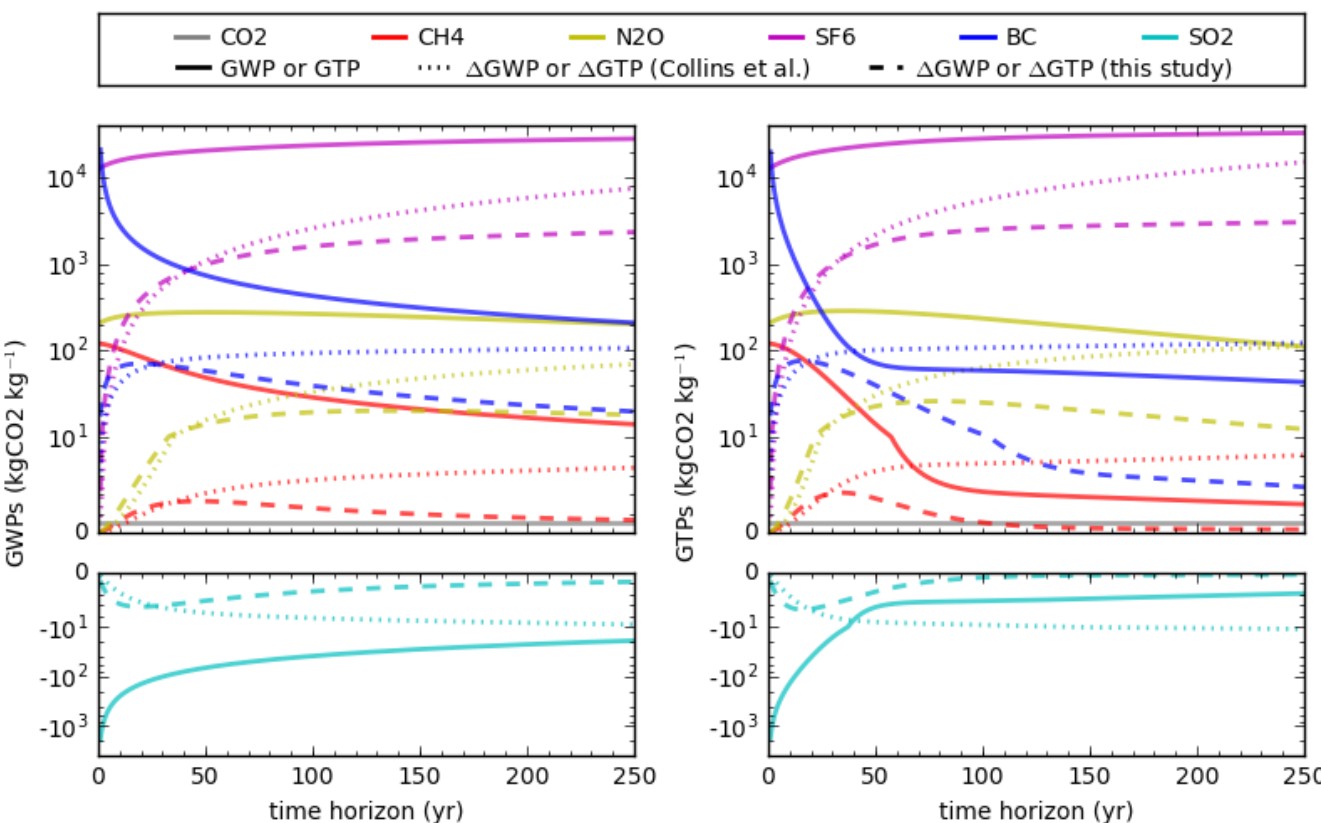

**Figure 6: ~~Normalized~~ Relative metrics, in the case of CH₄, N₂O, SF₆, BC and SO₂.** GWPs (left-hand side) and GTPs (right-hand side) obtained using the IPCC AR5 method are shown in solid lines. ΔGWPs and ΔGTPs obtained using the climate-carbon feedback IRF by Collins et al. (2013) are shown in dotted lines, and those obtained using ours are in dashed lines. Colours refer to the different species taken here as examples. Note that the scale of the *y*-axis is linear between 1 and ±10 and logarithmic afterwards.