# Peer review of "Accounting for the climate-carbon feedback in emission metrics"

_Earth System Dynamics, 2016_

## Short Comment (SC1) · 5 Dec 2016

This paper is a well-written, carefully constructed, and valuable contribution to the metrics literature. This paper improves upon the existing approach of incorporating climate-carbon feedbacks into GWP calculations and will constitute a very useful resource for subsequent assessments that will update these climate metrics. However, we strongly recommend the authors reconsider their recommendation to use the version of the GWP calculated with the climate-carbon feedback as the primary metric.

As noted in the manuscript, AR4 inconsistently calculated GWP's by including climate-carbon feedbacks for $CO_2$ perturbations but not for non-$CO_2$ perturbations. This inconsistency was noted in AR5 which presented climate metrics both with and without climate-carbon feedbacks for the non-$CO_2$ perturbations (based on Arora et al., 2013

and Collins et al., 2013), and we agree with the authors that this inconsistency should be resolved. The use of a climate-carbon feedback that more realistically incorporates an eventual relaxation back to a prior equilibrium for a pulse of climate change, as is presented in this paper, is an improvement to the calculation of a GWP that includes climate-carbon feedbacks. However, one option that was overlooked in AR5, and is presented as a secondary option in this paper, is to remove the climate-carbon feedbacks from both the CO2 and non-CO2 perturbations. We argue that for reasons of simplicity and transparency, that removing the climate-carbon feedback entirely is preferable for calculating GWPs for the use of policymakers.

GWPs have found favor among the metrics community for two primary reasons: ease of computation and simplicity/transparency. Including climate-carbon feedbacks may partially negate both of these benefits of the metric. Without climate-carbon feedbacks, one can calculate the absolute GWP for any given gas (with a known lifetime and radiative forcing) in a simple two-step process. Relative GWP then requires only the use of a previously-calculated 4-exponential function for CO2. However, the calculation of an absolute GWP with climate-carbon feedbacks is apparently a 10 step process (see Figure 4). Including the climate-carbon feedbacks is also shown in this paper to require additional assumptions beyond gas lifetime and radiative efficiency (the only two parameters necessary for the calculation of a traditional GWP). Requiring this choice reduces both the simplicity and transparency inherent in the GWP. Inclusion of climate-carbon feedbacks makes the value of the GWP dependent on attributes of the model chosen – its climate sensitivity, its rate of ocean uptake of heat, and how the carbon cycle changes in response to warming. This kind of additional complexity has been noted as a drawback of the GTP in comparison to the GWP. Incidentally, as the last equation on page 6 shows, the GTP is now effectively a necessary step in calculating the GWP using the methodology in this paper. This approach also requires additional, somewhat arbitrary choices: for example, the authors chose a climate change step of $+0.2°C$ to be approximately consistent with the peak warming of a 100 GtC pulse of CO2 and the approach of Joos et al. (2013), though Figure 3 shows that for step sizes

smaller than 1 degree, this choice does not appear to have a large impact.

Rows 5 and 6 of Table 2 demonstrate that there is little added value in terms of accuracy for the GWP when including the climate-carbon feedback (the 100 year GWP is particularly important as the metric in the most widespread use). The difference between the GWP calculated with and without climate-carbon feedbacks is less than 2% for any the 5 substances considered for any timeframe. This small improvement in accuracy of the 100 year GWP comes at the cost of complexity and lack of transparency as previously discussed. Moreover, despite the good work done by the authors in this paper, it is unclear to what extent use of a different model, parameters, or other choices could lead to changes in this small adjustment to the GWP.

The authors do note that "our results raise the question of whether the climate-carbon feedback should be included in emission metrics", and yet, they "recommend using the metrics shown in this fourth row of Table 2, since they are the most consistent, robust and up-to-date metrics available" (and even raise questions about what other feedbacks should be included, such as climate-wetlands feedbacks). We would strongly recommend that the authors reconsider this recommendation. The authors should continue to present the most up-to-date metrics based on their carbon-cycle models, but we believe that the authors should in fact endorse the use of the GWP without the climate-carbon feedback (in either the numerator or denominator). The authors show that including the climate-carbon feedback offers a slight improvement in accuracy, but in our opinion, that improvement in accuracy is far outweighed by the double drawbacks of increased difficulty of computation and loss of simplicity and transparency. Recommending the use of metrics without the feedback would mean highlighting the 3rd equation in Appendix C.1, as it would then be possible to calculate the GWP for any arbitrary non-CO2 gas given lifetime and radiative efficiency values. The authors could encourage other carbon-cycle modelers to similarly calculate carbon dioxide response functions without the inclusion of carbon-climate feedbacks such that a future IPCC assessment could draw from multiple studies to inform its GWP calculations.

We are, however, less opposed to the inclusion of climate-carbon feedbacks in the calculation of GTPs. Including the feedback in GTPs results in larger impacts than when considering the GWPs (almost 30% as the authors note, for the case of 20 year GTPs and either BC or SO2). Additionally, the additional computational cost, increase in complexity, and loss of transparency are much less powerful arguments when applied to the GTP in contrast to the GWP, since many of those drawbacks are inherent in GTP calculations in the first place.

Again, we commend the authors on an extremely interesting, robust, elegant, and useful analysis, but ask the authors to take our comments into account.

Thank you,

Marcus Sarofim (US EPA, Climate Change Division)

Michael Giordano (AAAS Science & Technology Policy Fellow placed at the US EPA)

Allison Crimmins (US EPA, Climate Change Division)

The views expressed in this comment are those of the authors and do not necessarily reflect the views or policies of the U.S. Environmental Protection Agency or other government agency.

Additional technical comments:

We note as a relevant comparison, that Reisinger et al. (2011) calculated the effect on GWPs of using different RCPs to project future concentrations, and found that GWPs could change by 10 to 30% for N2O, -10 to 20% for CH4, and 2 to 36% for the halocarbons. However, the community has to date retained the assumption of constant background concentration, presumably in order to preserve simplicity and to avoid the necessity of choosing a single future emissions scenario (or combination of scenarios). The effect of this simplifying assumption is an order of magnitude larger than that resulting from the inclusion of climate-carbon feedbacks.

[Figure]

Page 1, line 14: "the IPCC presented tentative values": The text of AR5 was not clear that the climate-carbon feedback values were to be considered "tentative", nor does this match the way that the AR5 values have been perceived and used by the community.

Page 2, line 16: "The standard metrics provided in the fifth assessment report": Similarly, the text of AR5 was not clear that the version of the metric that did not include climate-carbon feedbacks for non-CO2 gases (and was therefore "inconsistent"), should have been considered "the standard metric".

Page 11, line 16: The phrasing of the following sentence could be improved: "which is itself the value chosen by Joos et al. (2013) – and therefore by the IPCC": while the IPCC implicitly endorsed the approach of Joos et al., there was no explicit determination that 100 GtC or +0.2 degrees C is the optimal value to use. The IPCC can be limited by the literature available, and the choice of a given paper to support a parameter choice does not necessarily indicate endorsement of all the choices made within that paper. A preferable phrasing would be, "which is itself the value chosen by Joos et al (2013), which provided the carbon lifetime used by the IPCC" or something along those lines.

Page 11, line 18: the text refers to "Figure 5" as an illustration of the GTP calculation process, but should be corrected to refer to Figure 4.

Page 14, line 6: the authors do note that the inclusion of the feedback has less than a 10% impact on GWPs and GTPs, but the fact that the impact is less than 2% for other GWPs, even for short-lived species, for any time horizon, is an important distinction that is not sufficiently emphasized in the text. GTPs and GWPs are clearly impacted very differently here.

---

## Referee Comment (RC1) · Anonymous Referee #1 · 16 Dec 2016

This manuscript presents a methodology to better assess the greenhouse-gases emission metrics, by considering and removing the "climate-carbon" feedback that is implicitly used in previous estimations and in previous IPCC recommendations. The methodology is well exposed and rather straightforward, the scientific discussion is clear and well written. Therefore, I have no comment on the technical content of this paper. In contrast, I have some major comments on the overall presentation, introduction and conclusion : these critical comments must be accounted for by the authors before considering publication. Indeed, greenhouse-gases emission metrics is a very "subjective" tool that should be presented as such. It is possible to build a very accurate subjective methodology, but this certainly does not help to provide an objective one. I therefore strongly disagree with the general tone of the paper, given in the introduction:

page 1, line 20 : "However, including carbon-climate feedbacks, particularly in absolute

metrics or for short time horizons, gives a more realistic representation of the response"

I also strongly disagree with the conclusion that:

page 15, line 15: "To avoid potential biases in metric values, we suggest to include the climate-carbon feedback in metric estimates".

The very concept of a unique simple metric for GHG is both UNREALISTIC and BIASED. Refining this concept will not change this fundamental fact.

The purpose of GWPs or GTPs is to provide a unique simple metric to compare the "climatic impact" of the many different anthropogenic greenhouse gases (GHG). Obviously, from a scientific perspective, this amounts to comparing oranges and apples. I understand that such an exercise is necessary from a policy perspective, and that scientists should help and provide numbers. Still, I am not convinced that comparing "very accurately" oranges and apples is either necessary or desirable. At the very least, when comparing them, scientists should keep insisting on the differences.

The most important (and arbitrary) parameter is the chosen time horizon : do we value more the current generation (20 years from now) or future generations (500 years from now) ? This is a moral question, not a scientific one. Therefore, in the 2001 IPCC report, we read, for instance concerning methane ($CH_4$), a range of values: GWP20 = 62 ; GWP100 = 23 ; GWP500 = 7 (IPCC 2001, page 388, Table 6.7) Interestingly, the range given in the 2014 IPCC report (AR5) is "narrower": GWP20 = 84 ; GWP100 = 28 (IPCC 2014, page 731, Table 8.A.1) which does not reflect scientific advances or a more accurate assessment of the metric, but simply a different a priori choice, with the 500-year horizon not being discussed anymore in the last AR5 report. Similarly, using the GTP metric (the effect at final time t) instead of GWP (the effect integrated between gas injection and time t) is a rather arbitrary choice. The use of the global mean temperature (in GTPs), or global mean radiative forcing (in GWPs), is also quite arbitrary, since local impacts do not necessary scale linearly to such global averages. Of course, all these points have been discussed in the literature many times and are

well known to specialists. Still, I believe they are so critical and so often overlooked by non-specialists (policymakers, BUT also many climate scientists), than they need to be heavily stressed in papers on GHG metrics like the current manuscript. In particular, the reassessment of GWPs (or GTPs) performed in this manuscript, in order to "remove the carbon-cycle feedback in the denominator", does change the numerical values by, typically, a few percent or less, something very much smaller than, for instance, the arbitrary choice of a time horizon. This needs to be explicitly stated and probably strongly emphasized in the manuscript : comparing GHGs is much more a moral and subjective choice (eg. long-term versus short-term) than a scientific question. Providing accurate estimations of a subjective metric does not lead to an objective metric.

The very concept of GWPs/GTPs is based on a simple linear view of the climate system (impulse response functions, transfer functions, Laplace transforms, . . .). In order to be physically relevant, it requires the quite strong assumption that there is NO feedback at all in the system (ie. GWPs are fully independent on climate or other GHG levels). Of course, GWPs/GTPs can be diagnosed from complex non-linear systems, but their use as a simple metric is based on the assumption that the climate responds linearly to each individual forcing. The aim of the paper is therefore to remove the feedbacks in the carbon cycle to better "fit" into the concept of linear systems and therefore provide a more "accurate" quantification of GWPs/GTPs. But at the same time, climatologists usually insist in describing climate as a complex non-linear system, with many feedbacks (in particular between climate and the carbon cycle, precisely the one discussed in the paper). This is a point that deserves some extended discussion: To what extent GWPs/GTPs are sound concepts for climate? And to what extent are they simply imperfect tools designed to answer the heavy policy requirement for a metric ?

I have also a more specific problem with the IRF for temperature. The impulse response functions for carbon (Appendix C.1) have all the same structure: a constant term (= percent carbon staying in the atmosphere "forever") and several decreasing exponentials

(= capture of carbon by vegetation and ocean). In contrast, the impulse response functions used for temperature (Appendix C.2) have no constant term. In other words, a basic fundamental ASSUMPTION in the GTP computations is that climate change is fully reversible: whatever the size of the initial radiative "spike" forcing at time zero, climate recovers to its initial conditions after a few centuries. I have some major difficulties to admit such a strong HYPOTHESIS, which stands against all my knowledge in climate science... These response functions are obtained from atmosphere-ocean only GCMs simulations (without feedbacks from the surface vegetation changes, land ice cover, deep ocean changes, etc...) by fitting one-way experiments (abrupt or gradual 4xCO2 experiments with stabilization). Is this supposed to be realistic? Interestingly, there are no reversed experiments, even though the IRF functional form assumes reversibility: is this climate reversibility assumption based on something else than just simple convenience?

Again, I understand the requirement for a metric to compare GHG. Obviously, this implies some arbitrary choices and some drastic simplifications of the climate system. Still, I have difficulties with the logic of "fitting" the climate system into a simple linear fully reversible system. I certainly do not share the scientific concept behind. At the very least, these fundamental assumptions should be explicitly stated and discussed in the manuscript.

There is a real danger to misrepresent the response of the climatic system, in a "very accurate" BUT certainly not "objective" fashion, as a linear response to the superposition of independent GHG forcings that are not allowed to interact with each other, nor with climate. I am not sure that scientists should blindly misrepresent the real world, only to fit policy requirements of a simple metric. At least, they should be extremely cautious and stress the limitations of the GWPs/GTPs concept.

I am not a specialist of GHG metrics. I am writing this review just after the interactive comment from M. Sarofim et al. was posted online, and I strongly agree with it. The added value of a more "accurate" assessment of GWPs/GTPs, as presented in this

manuscript, comes at the cost of simplicity and reproducibility. Though the scientific methodology presented in this paper is sound and well presented, I am not sure this is the best way to fulfill the requirements of GHG metrics. Fundamentally, GHG metrics are only a "rule of thumb" to decide which GHG is "better" or "worse", from some subjective perspective. Scientists should not try to disguise this "rule of thumb" into an objective, quantified, assessment.

———————————————

---

## Short Comment (SC2) · 19 Dec 2016

Gasser et al. report on findings that the effect of including climate-carbon feedbacks for both the target species and CO2 produce GTP and GWP values that are much closer to their default values than was suggested in IPCC AR5 report. They also call for discussion in the community about the limits of the prevailing linear, impulse response function framework for describing complex feedbacks in the climate system.

I wanted to add some minor comments on the wording and equations:

Page 3 / L8: 'dynamic' – IRF describes a dynamic system, but not sure it is correct to say that it is dynamic; i.e. the impulse response functions are invariant with regards to initial time

Page 3: "the change in atmospheric concentration of the species (Qx)" - should be Qx(t) - Qx(0), or else equation on next line could simply be Qx(t) = . . .

Similar for the line about Tx

Page 4/L7: should be approximate ($\approx$) symbol rather than definition ($\equiv$)

Page 4: 'mass' would be more clear than 'size' in describing the emissions

The word 'normalized' and 'relative' are used inconsistently (it is not always used to say that GWP is relative while AGWP is absolute and that both are normalized (to a 1 kg pulse) metrics). This is apparent on page 4 / L9-11, but also applies page 2 / L18 and elsewhere

There is also a subtle change in notation that is not mentioned that some of the equations on page 5 and 6 give terms that were previously explicit functions of time but are now shown with implicit dependence through the variables Tx, Qx, E0x, RFx and $\Theta$

Page 5 / L26: a(t) should be a(t')

Page 6, L18: This seems to be the relation between pulse and continuous emission given in Aamaas et al. 2013, ESD 4: 145-170, but I could not follow the logic here. Also unclear that the 'definition' ($\equiv$) symbol is applicable

Page 9 / L19: the use of the word 'extended' causes a little confusion since the meaning is not described until L27-L29

Constant intensity term ($\lambda$) "climate sensitivity" : overall there was not much discussion of this parameter, but believe should at least point out that it refers to an equilibrium climate sensitivity

---

## Referee Comment (RC2) · KT Tanaka (Referee) · 19 Jan 2017

The authors present a new and elegant approach to including climate-carbon cycle feedbacks consistently in the estimates of emission metrics, and more specifically, absolute metrics for non-CO2 components. The paper makes several important points associated with the treatment of climate-carbon cycle feedbacks in the calculations of emission metrics performed for IPCC AR5. The text requires some editing (although I like the style of writing), but the argument is clear and the results are well presented. I think this paper potentially has a strong impact in the field of emission metrics and may influence the next IPCC report but can also lead to confusion among metric users as I discuss below. The paper requires a revision by reflecting the comments below before being recommended for publication in Earth System Dynamics.

[Figure]

I start with one broad comment, followed by several minor ones. The paper begins with the issue that the treatment of climate-carbon cycle feedbacks was inconsistent in representative metric values in IPCC AR5 (i.e. Table 8.A). More precisely, such feedbacks are accounted for in the estimates of absolute metrics for $CO_2$ but ignored in those for non-$CO_2$ components, resulting in an inconsistency when they are put together to calculate relative metrics. This inconsistency is, to be sure, clearly indicated in multiple places in IPCC AR5, but my observation is that the inconsistency has created confusion among metric users. Some studies that follow (e.g. (Cherubini et al. 2016; Levasseur et al. 2016)) support a use of alternative metric values taking climate-carbon cycle feedbacks consistently into account (i.e. Table 8.SM.15 in the Supplementary Material of IPCC AR5), even though alternative values are available only for a subset of the components of interest. Now, the paper reveals that the approach to incorporating climate-carbon cycle feedbacks for non-$CO_2$ components adopted in IPCC AR5 was actually wrong because the natural carbon sinks are assumed inactive for the additional $CO_2$ release through climate-carbon cycle feedbacks (e.g. Figure 2). This finding essentially disqualifies all the alternative metric values in IPCC AR5.

Given the situation above in the recent past, this paper may create a new confusion among metric users dealing with climate and environmental policies and assessments. I would therefore request a more detailed clarification of what has happened and what should be done for the metric values in IPCC AR5 in their view. I think that this paper is a right place to do so because some of the authors have been closely involved in the writing of the metric section of IPCC AR5.

Hopefully this comment can be taken in a constructive way, but the paper can be more explicit about why the treatment of climate-carbon cycle feedbacks ended up with being inconsistent in IPCC AR5. The paper describes how it is inconsistent in sufficient details (e.g. Page 5, Lines 3-9), but it is unclear to me why this has happened. For instance, why was it not possible to estimate an IRF for $CO_2$ response without climate-carbon cycle feedbacks? If this were available, this might have allowed one to estimate

metrics 'consistently' without climate-carbon cycle feedbacks. This might have been an alternative solution, if not a best one, in light of the inherent linear limitation in the IRF approach that is discussed in Section 5.2. In practice, it is probably not feasible to re-do an experiment requiring many models. But, looking back, was there a lack of coordination at the beginning? Furthermore, what about the method to account for climate-carbon cycle feedbacks for non-CO2 components in AR5? This method has not been sufficiently tested before the adoption and is based just on a section of one peer-reviewed paper (Collins et al. 2013), whose main contributions lie elsewhere. How was the ad-hoc decision process leading to the adoption of this approach made? Is there anything useful that can be learned from for the next IPCC report? What are the recommendations for metric values? I noticed that the paper does contain some text recommending the new estimates (page 12, lines 15-17), but it is buried in the middle of the paper and I am not sure what are the intentions. I raised some of the questions that may arise if the paper is officially published, although not all of them may not have to be answered in this paper. Clarifications suggested here should be helpful for metric applications, and ultimately the IPCC AR6.

Minor comments

Page 1, Lines 11-14 The argument concerns only a set of works using IRF to estimate emission metrics. There are also a body of relevant works based on other approaches like simple climate models (e.g. (Tanaka et al. 2009)) and more complicated ones (e.g. (Gillett and Matthews 2010)). These particular studies do consider climate-carbon cycle feedbacks to calculate emission metrics. The statement can be revised to be more restrictive.

Page 2, Line 2 Another area that I could think of is the ecosystem community (e.g. (Neubauer and Megonigal 2015)).

Page 2, Line 13 If there is any reference to support this statement for the last century, please add.

Page 2, Line 21 "whose" instead of "which"?

Page 3, Line 9 I don't think the underlying models exhibit a hysteresis within the range of IRF calibrations.

Page 4, Line 6 In practice, this pulse emission is large. As in Appendix A, it is 100 GtC in the case of CO2.

Page 5, Line 26 Should it be a(t') instead of a(t) in the integral?

Page 7, Line 16 It would be helpful if the authors provide a few sentences on how climate-carbon cycle feedbacks are modeled in OSCAR, rather than just a reference. Do the feedbacks act only on soil carbon? What about NPP? Do they directly affect the ocean carbon uptake?

Page 10, Line 3 Please elaborate on how this equation was derived.

Page 10, Lines 9-10 There are many earth system processes that are nonlinear. As something that has been discussed intensively before, I would point out the buffering of ocean CO2 uptake under rising atmospheric CO2 concentration. But this nonlinearity can be modeled by a revised IRF approach that treats the atmosphere and the mixed layer as one box (Hooss et al. 2001).

Page 11, Lines 4-5 Related to the comment above, the authors should refer to the relevant debate on the linear limitation of IRF (Joos et al. 1996; Hooss et al. 2001). A detailed biogeochemical discussion is given in Section 2.1.2 of (Tanaka et al. 2007).

Page 12, Line 2 This should be "figures 5 and 6" because there is no figure 7 in the current manuscript.

Page 12, Lines 15-17 Related to my major comment, if this is really a recommendation for metric users, this needs to be more highlighted in the text. Metric users would otherwise be left wonder what are the values that should be used for applications.

Page 13, Line 20 This is just a minor note, but TOTEM (Ver et al. 1999; Mackenzie et
al. 2011), which is one of the models used to derive the IPCC AR5 IRF (Joos et al. 2013), accounts for nitrogen and phosphorus limitations.

Page 14, Lines 29-30 I fully agree with this statement.

Page 15, Lines 16-17 I am coming back to the first minor comment. Although I somewhat hesitate to repeat this point because of the conflict of interest, the paper should discuss studies that estimate emission metrics based on models other than IRFs at least at some length. Examples are (Manne and Richels 2001; Tanaka et al. 2009; Gillett and Matthews 2010; Reisinger et al. 2010; Johansson 2012; Smith et al. 2012; Tanaka et al. 2013; Sterner et al. 2014), and there are many more. The current manuscript narrowly focuses on IRF-based studies. I believe that adding more relevant studies should enrich the discussion in this paper and make the argument more convincing.

Page 27 Figure 4 is not discussed in the paper.

References

Cherubini F, Fuglestvedt J, Gasser T, Reisinger A, Cavalett O, Huijbregts MAJ, Johansson DJA, Jørgensen SV, Raugei M, Schivley G, Strømman AH, Tanaka K, Levasseur A (2016) Bridging the gap between impact assessment methods and climate science. Environmental Science & Policy 64:129-140. doi:10.1016/j.envsci.2016.06.019

Collins WJ, Fry MM, Yu H, Fuglestvedt JS, Shindell DT, West JJ (2013) Global and regional temperature-change potentials for near-term climate forcers. Atmos Chem Phys 13 (5):2471-2485. doi:10.5194/acp-13-2471-2013

Gillett NP, Matthews HD (2010) Accounting for carbon cycle feedbacks in a comparison of the global warming effects of greenhouse gases. Environ Res Lett 5 (3):034011. doi:10.1088/1748-9326/5/3/034011

Hooss G, Voss R, Hasselmann K, Maier-Reimer E, Joos F (2001) A nonlinear impulse response model of the coupled carbon cycle-climate system (NICCS). Clim Dyn 18

(3):189-202. doi:10.1007/s003820100170

Johansson D (2012) Economics- and physical-based metrics for comparing greenhouse gases. Clim Change 110 (1):123-141. doi:10.1007/s10584-011-0072-2

Joos F, Bruno M, Fink R, Siegenthaler U, Stocker TF, Le Quélé C, Sarmiento JL (1996) An efficient and accurate representation of complex oceanic and biospheric models of anthropogenic carbon uptake. Tellus B 48:397-417. doi:10.1034/j.1600-0889.1996.t01-2-00006.x

Joos F, Roth R, Fuglestvedt JS, Peters GP, Enting IG, von Bloh W, Brovkin V, Burke EJ, Eby M, Edwards NR, Friedrich T, Frölicher TL, Halloran PR, Holden PB, Jones C, Kleinen T, Mackenzie FT, Matsumoto K, Meinshausen M, Plattner G-K, Reisinger A, Segschneider J, Shaffer G, Steinacher M, Strassmann K, Tanaka K, Timmermann A, Weaver AJ (2013) Carbon dioxide and climate impulse response functions for the computation of greenhouse gas metrics: a multi-model analysis. Atmospheric Chemistry and Physics 13 (5):2793-2825. doi:10.5194/acp-13-2793-2013

Levasseur A, Cavalett O, Fuglestvedt JS, Gasser T, Johansson DJA, Jørgensen SV, Raugei M, Reisinger A, Schivley G, Strømman A, Tanaka K, Cherubini F (2016) Enhancing life cycle impact assessment from climate science: Review of recent findings and recommendations for application to LCA. Ecol Indicators 71:163-174. doi:10.1016/j.ecolind.2016.06.049

Mackenzie FT, De Carlo EH, Lerman A (2011) Coupled C, N, P, and O biogeochemical cycling at the land-ocean interface. In: Wolanski E, McLusky DS (eds) Treatise on Estuarine and Coastal Science, vol 5. Academic Press, Waltham, pp 317-342

Manne AS, Richels RG (2001) An alternative approach to establishing trade-offs among greenhouse gases. Nature 410 (6829):675-677

Neubauer SC, Megonigal JP (2015) Moving Beyond Global Warming Potentials to Quantify the Climatic Role of Ecosystems. Ecosystems 18 (6):1000-1013.
doi:10.1007/s10021-015-9879-4

Reisinger A, Meinshausen M, Manning M, Bodeker G (2010) Uncertainties of global warming metrics: CO2 and CH4. Geophys Res Lett 37 (14):L14707. doi:10.1029/2010gl043803

Smith SM, Lowe JA, Bowerman NHA, Gohar LK, Huntingford C, Allen MR (2012) Equivalence of greenhouse-gas emissions for peak temperature limits. Nature Clim Change 2 (7):535-538. doi:org/10.1038/nclimate1496

Sterner E, Johansson DJA, Azar C (2014) Emission metrics and sea level rise. Clim Change 127 (2):335-351. doi:10.1007/s10584-014-1258-1

Tanaka K, Johansson DJA, O'Neill BC, Fuglestvedt JS (2013) Emission metrics under the 2°C climate stabilization target. Clim Change 117:933-941. doi:10.1007/s10584-013-0693-8

Tanaka K, Kriegler E, Bruckner T, Hooss G, Knorr W, Raddatz T (2007) Aggregated Carbon Cycle, Atmospheric Chemistry, and Climate Model (ACC2) – description of the forward and inverse modes. Reports on Earth System Science, vol 40. Max Planck Institute for Meteorology, Hamburg

Tanaka K, O'Neill BC, Rokityanskiy D, Obersteiner M, Tol R (2009) Evaluating Global Warming Potentials with historical temperature. Clim Change 96 (4):443-466. doi:10.1007/s10584-009-9566-6

Ver LMB, Mackenzie FT, Lerman A (1999) Biogeochemical responses of the carbon cycle to natural and human perturbations: Past, present, and future. Am J Sci 299 (7-9):762-801

---

## Author Comment (AC1) · 2 Mar 2017

This paper is a well-written, carefully constructed, and valuable contribution to the metrics literature. This paper improves upon the existing approach of incorporating climate-carbon feedbacks into GWP calculations and will constitute a very useful resource for subsequent assessments that will update these climate metrics. However, we strongly recommend the authors reconsider their recommendation to use the version of the GWP calculated with the climate-carbon feedback as the primary metric.

Thank you for your support.

As noted in the manuscript, AR4 inconsistently calculated GWP's by including climate-carbon feedbacks for $CO_2$ perturbations but not for non-$CO_2$ perturbations. This in-

consistency was noted in AR5 which presented climate metrics both with and without climate-carbon feedbacks for the non-CO2 perturbations (based on Arora et al., 2013 and Collins et al., 2013), and we agree with the authors that this inconsistency should be resolved. The use of a climate-carbon feedback that more realistically incorporates an eventual relaxation back to a prior equilibrium for a pulse of climate change, as is presented in this paper, is an improvement to the calculation of a GWP that includes climate-carbon feedbacks. However, one option that was overlooked in AR5, and is presented as a secondary option in this paper, is to remove the climate-carbon feedbacks from both the CO2 and non-CO2 perturbations. We argue that for reasons of simplicity and transparency, that removing the climate-carbon feedback entirely is preferable for calculating GWPs for the use of policymakers.

GWPs have found favor among the metrics community for two primary reasons: ease of computation and simplicity/transparency. Including climate-carbon feedbacks may partially negate both of these benefits of the metric. Without climate-carbon feedbacks, one can calculate the absolute GWP for any given gas (with a known lifetime and radiative forcing) in a simple two-step process. Relative GWP then requires only the use of a previously-calculated 4-exponential function for CO2. However, the calculation of an absolute GWP with climate-carbon feedbacks is apparently a 10 step process (see Figure 4). Including the climate-carbon feedbacks is also shown in this paper to require additional assumptions beyond gas lifetime and radiative efficiency (the only two parameters necessary for the calculation of a traditional GWP). Requiring this choice reduces both the simplicity and transparency inherent in the GWP. Inclusion of climate-carbon feedbacks makes the value of the GWP dependent on attributes of the model chosen – its climate sensitivity, its rate of ocean uptake of heat, and how the carbon cycle changes in response to warming. This kind of additional complexity has been noted as a drawback of the GTP in comparison to the GWP. Incidentally, as the last equation on page 6 shows, the GTP is now effectively a necessary step in calculating the GWP using the methodology in this paper. This approach also requires additional, somewhat arbitrary choices: for example, the authors chose a climate change step of

+0.2 degC to be approximately consistent with the peak warming of a 100 GtC pulse of $CO_2$ and the approach of Joos et al. (2013), though Figure 3 shows that for step sizes smaller than 1 degree, this choice does not appear to have a large impact.

Rows 5 and 6 of Table 2 demonstrate that there is little added value in terms of accuracy for the GWP when including the climate-carbon feedback (the 100 year GWP is particularly important as the metric in the most widespread use). The difference between the GWP calculated with and without climate-carbon feedbacks is less than 2% for any the 5 substances considered for any timeframe. This small improvement in accuracy of the 100 year GWP comes at the cost of complexity and lack of transparency as previously discussed. Moreover, despite the good work done by the authors in this paper, it is unclear to what extent use of a different model, parameters, or other choices could lead to changes in this small adjustment to the GWP.

The authors do note that "our results raise the question of whether the climate-carbon feedback should be included in emission metrics", and yet, they "recommend using the metrics shown in this fourth row of Table 2, since they are the most consistent, robust and up-to-date metrics available" (and even raise questions about what other feedbacks should be included, such as climate-wetlands feedbacks). We would strongly recommend that the authors reconsider this recommendation.

The authors should continue to present the most up-to-date metrics based on their carbon-cycle models, but we believe that the authors should in fact endorse the use of the GWP without the climate-carbon feedback (in either the numerator or denominator). The authors show that including the climate-carbon feedback offers a slight improvement in accuracy, but in our opinion, that improvement in accuracy is far outweighed by the double drawbacks of increased difficulty of computation and loss of simplicity and transparency.

Recommending the use of metrics without the feedback would mean highlighting the 3rd equation in Appendix C.1, as it would then be possible to calculate the GWP for any

arbitrary non-CO2 gas given lifetime and radiative efficiency values. The authors could encourage other carbon-cycle modelers to similarly calculate carbon dioxide response functions without the inclusion of carbon-climate feedbacks such that a future IPCC assessment could draw from multiple studies to inform its GWP calculations.

We are, however, less opposed to the inclusion of climate-carbon feedbacks in the calculation of GTPs. Including the feedback in GTPs results in larger impacts than when considering the GWPs (almost 30% as the authors note, for the case of 20 year GTPs and either BC or SO2). Additionally, the additional computational cost, increase in complexity, and loss of transparency are much less powerful arguments when applied to the GTP in contrast to the GWP, since many of those drawbacks are inherent in GTP calculations in the first place.

Again, we commend the authors on an extremely interesting, robust, elegant, and useful analysis, but ask the authors to take our comments into account.

This comment makes a very strong case against including the feedback in GWPs. And we do agree with most of the points made. This is actually a debate we had, amongst authors, and in the first submitted version of the paper we settled on recommending including the feedback because we also had the absolute metrics in mind. Sometimes, AGWPs and AGTPs are used as very simple models of climate change. In this case, including the feedback improves the accuracy of the representation of the system.

However, we have decided that in the revised version there will be no recommendation at all. Partly because we cannot clearly recommend one of the two options, but mostly because we now think that making any recommendation would make this paper too controversial, and would weaken its more scientific/methodological aspects.

We initially wrote the paper as a research paper. And though we understand that a part of the community is eagerly waiting for a recommendation and consistently updated metrics, we believe our paper is not the place where this should happen. A commentary or perspective paper could follow, of course, but we are not convinced it

is our role to lead such a paper (although we can surely contribute). As the sought recommendations ultimately concern more the user community, maybe this community should be the one making those recommendations.

So in concrete terms, in the text, we have removed all previous recommendations, and rewritten the first paragraph of section 5.2:

"In the case of absolute metrics – both AGWPs and AGTPs – these changes are substantial since we are adding a positive feedback to the model. Therefore, the choice of including or excluding the feedback may ultimately depend on the application of the metrics. On the one hand, for the sake of simplicity and transparency, the feedback could be excluded from the evaluation of GWPs, since it avoids the trouble of the five convolutions shown in figure 4. On the other hand, if absolute (e.g. time-varying) metrics are used as a first-order model of climate change, one may prefer including the climate-carbon feedback to have a better representation of the system."

and a part of the conclusion:

"Ultimately, whether emission metrics should include the climate-carbon feedback is a decision for the user, and we only recommend consistency in the way feedbacks are included or excluded. The trade-off between simplicity and transparency on the one hand, and accuracy of representation on the other hand, has to be weighed by the final user."

We note as a relevant comparison, that Reisinger et al. (2011) calculated the effect on GWPs of using different RCPs to project future concentrations, and found that GWPs could change by 10 to 30% for N2O, -10 to 20% for CH4, and 2 to 36% for the halocarbons. However, the community has to date retained the assumption of constant background concentration, presumably in order to preserve simplicity and to avoid the necessity of choosing a single future emissions scenario (or combination of scenarios). The effect of this simplifying assumption is an order of magnitude larger than that
resulting from the inclusion of climate-carbon feedbacks.

Yes, just as we point out that the background strongly affects the calibrated IRF (and subsequent metrics). We've added a sentence to note the larger effect of the background: "[Variation in the metrics' value from including/excluding the feedback] are also less in magnitude than those induced by the choice of the protocol used to calculate the metrics, such as the background conditions (e.g. Reisinger et al., 2011), or by the choice of a given time horizon (see e.g. table 2)."

Page 1, line 14: "the IPCC presented tentative values": The text of AR5 was not clear that the climate-carbon feedback values were to be considered "tentative", nor does this match the way that the AR5 values have been perceived and used by the community.

Removed. (But see next point.)

Page 2, line 16: "The standard metrics provided in the fifth assessment report": Similarly, the text of AR5 was not clear that the version of the metric that did not include climate-carbon feedbacks for non-CO2 gases (and was therefore "inconsistent"), should have been considered "the standard metric".

We disagree with this comment: the huge table at the end of chapter 8 of IPCC is the main product of this section of the report.

However, we agree that the fact that it was published as an appendix (because of its size) made it difficult for the metric-user community to perceive which of the two tables (w/ & w/o feedback) was the main product...

Page 11, line 16: The phrasing of the following sentence could be improved: "which is itself the value chosen by Joos et al. (2013) – and therefore by the IPCC": while the IPCC implicitly endorsed the approach of Joos et al., there was no explicit determination that 100 GtC or +0.2 degrees C is the optimal value to use. The IPCC can be limited by the literature available, and the choice of a given paper to support a parameter choice does not necessarily indicate endorsement of all the choices made within that paper. A preferable phrasing would be, "which is itself the value chosen by Joos et al (2013), which provided the carbon lifetime used by the IPCC" or something along those lines.

Agreed. The end of the sentence is now: "chosen by Joos et al. (2013) and used in the IPCC AR5."

Page 11, line 18: the text refers to "Figure 5" as an illustration of the GTP calculation process, but should be corrected to refer to Figure 4.

Yes, we have corrected the numbering.

Page 14, line 6: the authors do note that the inclusion of the feedback has less than a 10% impact on GWPs and GTPs, but the fact that the impact is less than 2% for other GWPs, even for short-lived species, for any time horizon, is an important distinction that is not sufficiently emphasized in the text. GTPs and GWPs are clearly impacted very differently here.

We have reformulated the discussion to more clearly separate the effect on GWPs and GTPs:

"We have found that including or excluding the climate-carbon feedback in a consistent manner does not greatly change the values of the relative GWPs (only about 2%). In the case of relative GTPs, the change is slightly larger for greenhouse gases (less than 10%) and becomes even larger for very short-lived species and over short time-horizons (greater than 30%)."

---

## Author Comment (AC2) · 2 Mar 2017

Gasser et al. report on findings that the effect of including climate-carbon feedbacks for both the target species and CO2 produce GTP and GWP values that are much closer to their default values than was suggested in IPCC AR5 report. They also call for discussion in the community about the limits of the prevailing linear, impulse response function framework for describing complex feedbacks in the climate system.

I wanted to add some minor comments on the wording and equations:

Thank you for the comments.

Page 3 / L8: 'dynamic' – IRF describes a dynamic system, but not sure it is correct to

say that it is dynamic; i.e. the impulse response functions are invariant with regards to initial time

An IRF is the analytical solution of a set of ordinary differential equations with constant coefficients (i.e. linear). Therefore, it describes the exact same (physical) system as the differential equations do. If the system is deemed dynamic, we see no reason why the IRF should not be said to be dynamic as well.

Page 3: "the change in atmospheric concentration of the species (Qx)" - should be Qx(t) - Qx(0), or else equation on next line could simply be Qx(t) = ... Similar for the line about Tx

Note that the sentence can be read slightly differently: "the change in [atmospheric concentration of the species (Qx)]"; i.e. "Qx" is attached only to "atmospheric concentration of the species" and not to the whole beginning of the sentence.

Moreover, if we assume the sentence is read as the commenter suggests, the equation has to be written with Qx(0) on the right-hand side. But this is completely equivalent to the way we wrote it.

So we don't think any change to the sentence should be made, firstly for the sake of legibility.

Page 4/L7: should be approximate ($\approx$) symbol rather than definition ($\equiv$)

This has been changed to: $RF^x(t) = \varphi^x(Q^x(t) - Q^x(0))$

Page 4: 'mass' would be more clear than 'size' in describing the emissions

Though 'mass' would be physically correct, we prefer to use the more colloquial and yet widely used word 'size', as Joos et al. (2013) and IPCC did.

The word 'normalized' and 'relative' are used inconsistently (it is not always used to say that GWP is relative while AGWP is absolute and that both are normalized (to a 1 kg pulse) metrics). This is apparent on page 4 / L9-11, but also applies page 2 / L18 and elsewhere

Very true! We checked and corrected all occurrences.

There is also a subtle change in notation that is not mentioned that some of the equations on page 5 and 6 give terms that were previously explicit functions of time but are now shown with implicit dependence through the variables Tx, Qx, E0x, RFx and Θ

The change in notation is explicitly introduced with the following (existing) sentence: "To simplify the discussion and avoid quintuple integrals, we introduce the simplified notation $\star$ for the convolution: $a \star b \equiv \int_0^t a(t')b(t-t')dt'$, and note the commutative property of the convolution: $a \star b = b \star a$."

The disappearance of the time variable comes with the new notation, and happens for the same reason: to keep the notation simple and legible.

Page 5 / L26: a(t) should be a(t')

Yes. Corrected.

Page 6, L18: This seems to be the relation between pulse and continuous emission given in Aamaas et al. 2013, ESD 4: 145-170, but I could not follow the logic here. Also unclear that the 'definition' ($\equiv$) symbol is applicable

We have added a short sentence to make this part clearer: "[Note] that convoluting any function with the Heaviside function is equivalent to integrating it."

Although, we have to say there is no 'logic' here apart from acknowledging that the

integral of a function f is – by definition – strictly equal to the convolution of f by the Heaviside function.

Page 9 / L19: the use of the word 'extended' causes a little confusion since the meaning is not described until L27-L29

Yes. We have moved that sentence to appendix B, where the details of the fit are given.

Constant intensity term ($\lambda$) "climate sensitivity": overall there was not much discussion of this parameter, but believe should at least point out that it refers to an equilibrium climate sensitivity.

The parameter is defined in section 2.1. We have added the word "equilibrium" to the text to make it clearer.

---

## Author Comment (AC3) · 2 Mar 2017

This manuscript presents a methodology to better assess the greenhouse-gases emission metrics, by considering and removing the "climate-carbon" feedback that is implicitly used in previous estimations and in previous IPCC recommendations. The methodology is well exposed and rather straightforward, the scientific discussion is clear and well written. Therefore, I have no comment on the technical content of this paper.

We thank the referee for acknowledging the technical quality of the paper.

In contrast, I have some major comments on the overall presentation, introduction and conclusion: these critical comments must be accounted for by the authors before considering publication. Indeed, greenhouse-gases emission metrics is a very "subjective"

tool that should be presented as such. It is possible to build a very accurate subjective methodology, but this certainly does not help to provide an objective one.

Nowhere in this paper do we pretend to create an "objective" metric. The subjective aspects of emission metrics have been largely discussed in the literature (e.g. IPCC AR5 WG1 Chapter 8), and the topic falls out of the scope of our paper. Here, we discuss scientific and technical issues regarding the inclusion of the climate-carbon feedback in metrics.

I therefore strongly disagree with the general tone of the paper, given in the introduction: page 1, line 20: "However, including carbon-climate feedbacks, particularly in absolute metrics or for short time horizons, gives a more realistic representation of the response"

This sentence has been changed: "Including or excluding the climate-carbon feedback ultimately depends on the user's goal, but consistency should be ensured in either case."

It now reflects the fact that we do not recommend a particular approach, between including and removing the feedback. We do recommend, however, a consistent approach, and therefore to update the IPCC metric estimates.

I also strongly disagree with the conclusion that: page 15, line 15: "To avoid potential biases in metric values, we suggest to include the climate-carbon feedback in metric estimates".

This has been changed as well. The concluding paragraph now is:

"Ultimately, whether emission metrics should include the climate-carbon feedback is a decision for the user, and we only recommend consistency in the way feedbacks are included or excluded. The trade-off between simplicity and transparency on the

one hand, and accuracy of representation on the other hand, has to be weighed by the final user. But metric users should also keep in mind that IRFs and emission metrics are extremely simple models of a complex system, and that sometimes it may be beneficial to use more complex models that better capture multiple and interacting feedback processes."

The very concept of a unique simple metric for GHG is both UNREALISTIC and BIASED. Refining this concept will not change this fundamental fact. The purpose of GWPs or GTPs is to provide a unique simple metric to compare the "climatic impact" of the many different anthropogenic greenhouse gases (GHG). Obviously, from a scientific perspective, this amounts to comparing oranges and apples. I understand that such an exercise is necessary from a policy perspective, and that scientists should help and provide numbers. Still, I am not convinced that comparing "very accurately" oranges and apples is either necessary or desirable. At the very least, when comparing them, scientists should keep insisting on the differences.

Our paper is embedded in the existing literature on emissions metrics. The basic premise of the paper was to reassess the way that feedbacks are included in metrics, an issue noted as requiring research (IPCC AR5 WG1 Chapter 8). As the reviewer notes, metrics may be "necessary from a policy perspective, and that scientists should help and provide numbers". We see this paper as fulfilling a request from policy makers, to show and correct a mistake made by the IPCC. We additionally show several metrics (GWP, GTP) and different time horizon, and show the importance of feedbacks. We do not recommend one metric over another; that is not our role.

The most important (and arbitrary) parameter is the chosen time horizon: do we value more the current generation (20 years from now) or future generations (500 years from now)? This is a moral question, not a scientific one. Therefore, in the 2001 IPCC report, we read, for instance concerning methane (CH4), a range of values: GWP20 =

62 ; GWP100 = 23 ; GWP500 = 7 (IPCC 2001, page 388, Table 6.7) Interestingly, the range given in the 2014 IPCC report (AR5) is "narrower": GWP20 = 84 ; GWP100 = 28 (IPCC 2014, page 731, Table 8.A.1) which does not reflect scientific advances or a more accurate assessment of the metric, but simply a different a priori choice, with the 500-year horizon not being discussed anymore in the last AR5 report. Similarly, using the GTP metric (the effect at final time t) instead of GWP (the effect integrated between gas injection and time t) is a rather arbitrary choice. The use of the global mean temperature (in GTPs), or global mean radiative forcing (in GWPs), is also quite arbitrary, since local impacts do not necessary scale linearly to such global averages. Of course, all these points have been discussed in the literature many times and are well known to specialists. Still, I believe they are so critical and so often overlooked by non-specialists (policymakers, BUT also many climate scientists), than they need to be heavily stressed in papers on GHG metrics like the current manuscript. In particular, the reassessment of GWPs (or GTPs) performed in this manuscript, in order to "remove the carbon-cycle feedback in the denominator", does change the numerical values by, typically, a few percent or less, something very much smaller than, for instance, the arbitrary choice of a time horizon. This needs to be explicitly stated and probably strongly emphasized in the manuscript: comparing GHGs is much more a moral and subjective choice (eg. long-term versus short-term) than a scientific question. Providing accurate estimations of a subjective metric does not lead to an objective metric.

We have added a sentence in the discussion/conclusion to recall that the time horizon remains an important choice when calculating emission metrics: "[Variation in the metrics' value from including/excluding the feedback] are also less in magnitude than those induced by the choice of the protocol used to calculate the metrics, such as the background conditions (e.g. Reisinger et al., 2011), or by the choice of a given time horizon (see e.g. table 2)".

The fact that there is a first order uncertainty does not prevent studying a second order one, especially since the first order uncertainty is of political nature whereas the second

order one is of scientific nature.

The very concept of GWPs/GTPs is based on a simple linear view of the climate system (impulse response functions, transfer functions, Laplace transforms, . . .). In order to be physically relevant, it requires the quite strong assumption that there is NO feedback at all in the system (ie. GWPs are fully independent on climate or other GHG levels). Of course, GWPs/GTPs can be diagnosed from complex non-linear systems, but their use as a simple metric is based on the assumption that the climate responds linearly to each individual forcing.

The original purpose of emissions metrics was to compare GHG emissions at the margin (e.g., IPCC FAR). The general concept is to compare one additional kg of different GHGs. In practice, because of the signal-to-noise ratio, large pulses are used to estimate IRFs (e.g. Joos et al., 2013). Though, test are performed to ensure the pulse is not so large as to introduce non-linear responses (e.g. Joos et al., 2013, but also our figure 3).

IRFs and metrics do include some types of feedbacks, with the strong limitation that they are implicitly and linearly accounted for. For instance, the climate IRF implicitly includes the water vapor, lapse rate, cloud-cover and sea-ice feedbacks.

Our paper also demonstrate that it is possible to account for more feedbacks by developing further the IRF framework. And as the work of Joos et al. (2013) shows, despite the feedbacks, it is still possible to look at a linear (marginal) response.

The aim of the paper is therefore to remove the feedbacks in the carbon cycle to better "fit" into the concept of linear systems and therefore provide a more "accurate" quantification of GWPs/GTPs.

This is not the goal of the paper. The first goal is to correct the IPCC mistake by making

the GWPs and GTPs consistent in the way they include the climate-carbon feedback. The second goal is to discuss how the metrics are affected by including or excluding the feedback.

But at the same time, climatologists usually insist in describing climate as a complex non-linear system, with many feedbacks (in particular between climate and the carbon cycle, precisely the one discussed in the paper). This is a point that deserves some extended discussion: To what extent GWPs/GTPs are sound concepts for climate? And to what extent are they simply imperfect tools designed to answer the heavy policy requirement for a metric?

The discussion suggested by the referee is way out of the scope of the paper. Our paper is not a review nor a perspective on the topic. It is based on the existing literature and solves one previously identified issue of the emission metrics.

Further, IRFs and metrics are designed to be used at the margin where linearity holds, and they are used here to compare pulses of GHGs (not emission scenarios where linearity becomes a problem).

I have also a more specific problem with the IRF for temperature. The impulse response functions for carbon (Appendix C.1) have all the same structure: a constant term (= percent carbon staying in the atmosphere "forever") and several decreasing exponentials (= capture of carbon by vegetation and ocean). In contrast, the impulse response functions used for temperature (Appendix C.2) have no constant term. In other words, a basic fundamental ASSUMPTION in the GTP computations is that climate change is fully reversible: whatever the size of the initial radiative "spike" forcing at time zero, climate recovers to its initial conditions after a few centuries. I have some major difficulties to admit such a strong HYPOTHESIS, which stands against all my knowledge in climate science... These response functions are obtained from atmosphere-ocean only

GCMs simulations (without feedbacks from the surface vegetation changes, land ice cover, deep ocean changes, etc...) by fitting one-way experiments (abrupt or gradual 4xCO2 experiments with stabilization). Is this supposed to be realistic? Interestingly, there are no reversed experiments, even though the IRF functional form assumes reversibility: is this climate reversibility assumption based on something else than just simple convenience?

Again, I understand the requirement for a metric to compare GHG. Obviously, this implies some arbitrary choices and some drastic simplifications of the climate system. Still, I have difficulties with the logic of "fitting" the climate system into a simple linear fully reversible system. I certainly do not share the scientific concept behind. At the very least, these fundamental assumptions should be explicitly stated and discussed in the manuscript.

It is true that full reversibility is implicitly assumed, so we have added the following in the introductory paragraph of the section dedicated to IRFs, to remind the reader of the implicit reversibility of the model: "[IRFs] represent a fully reversible system [...]". With IRFs, however, this reversibility is not instantaneous, and such a model is fully capable of showing the kind of hysteresis one can observe in complex models.

Here, in the specific case of emission metrics, the idea that those are calculated as the contribution of a marginal emission of the considered gas is also important. The marginal emission of the gas is assumed to occur under a given background, but it is not assumed to affect this background. Therefore, under the metric framework there is no issue of irreversibility. The issue only arises if one wants to use the IRFs as first-order models of the climate system, which is not the case in this paper.

Note also that the constant term in the carbon-cycle response is not a proper irreversibility. It is an apparent irreversibility within the time-frame of the experiment used to calibrate the IRF (1000 years). IRFs over a longer time-frame have been proposed, in which case the constant term becomes also a decaying exponential with a time-scale

much longer than 1000 years (about 80.000 years if we stick to only one exponential to describe this long-term response).

There is a real danger to misrepresent the response of the climatic system, in a "very accurate" BUT certainly not "objective" fashion, as a linear response to the superposition of independent GHG forcings that are not allowed to interact with each other, nor with climate. I am not sure that scientists should blindly misrepresent the real world, only to fit policy requirements of a simple metric. At least, they should be extremely cautious and stress the limitations of the GWPs/GTPs concept.

"Remember that all models are wrong; the practical question is how wrong do they have to be to not be useful." (Box G .E. P. & Draper N. R., 1987, Empirical Model Building and Response Surfaces).

Though the quote might be overused, it remains – we believe – a fundamental truth about modeling. We argue that the (political) demand for metrics such as GWPs and GTPs makes those useful de facto. We acknowledge the many limitations of those simplistic models. But these have been discussed extensively in the literature, and it is not within the scope of this paper to revisit the question.

I am not a specialist of GHG metrics. I am writing this review just after the interactive comment from M. Sarofim et al. was posted online, and I strongly agree with it. The added value of a more "accurate" assessment of GWPs/GTPs, as presented in this manuscript, comes at the cost of simplicity and reproducibility. Though the scientific methodology presented in this paper is sound and well presented, I am not sure this is the best way to fulfill the requirements of GHG metrics. Fundamentally, GHG metrics are only a "rule of thumb" to decide which GHG is "better" or "worse", from some subjective perspective. Scientists should not try to disguise this "rule of thumb" into an objective, quantified, assessment.

As stated by the reviewer, emission metrics are used to compare GHGs, to decide if one is better or worse than the other from some subjective perspective. Yes, that sums up an emission metric. But, policy makers have a need for such a tool (such as in emission trading). All the authors on this paper are fully aware of the limitations of emission metrics, and have written on the subjective aspects (some extensively). We are in no way trying to "disguise" this subjectivity. This paper is well embedded in the existing literature. The paper is of a technical nature and discusses a technical issue, and therefore readers would go elsewhere for a more detailed discussion of the subjective aspects of metrics (many of which we mention and cite). It is not in the scope of the paper, nor the interests of the readers, to discuss a topic that has been well discussed, reviewed, and assessed by others.

---

## Author Comment (AC4) · 2 Mar 2017

The authors present a new and elegant approach to including climate-carbon cycle feedbacks consistently in the estimates of emission metrics, and more specifically, absolute metrics for non-CO2 components. The paper makes several important points associated with the treatment of climate-carbon cycle feedbacks in the calculations of emission metrics performed for IPCC AR5. The text requires some editing (although I like the style of writing), but the argument is clear and the results are well presented. I think this paper potentially has a strong impact in the field of emission metrics and may influence the next IPCC report but can also lead to confusion among metric users as I discuss below. The paper requires a revision by reflecting the comments below before being recommended for publication in Earth System Dynamics.

We thank the referee for his review and support.

I start with one broad comment, followed by several minor ones. The paper begins with the issue that the treatment of climate-carbon cycle feedbacks was inconsistent in representative metric values in IPCC AR5 (i.e. Table 8.A). More precisely, such feedbacks are accounted for in the estimates of absolute metrics for $CO_2$ but ignored in those for non-$CO_2$ components, resulting in an inconsistency when they are put together to calculate relative metrics. This inconsistency is, to be sure, clearly indicated in multiple places in IPCC AR5, but my observation is that the inconsistency has created confusion among metric users. Some studies that follow (e.g. (Cherubini et al. 2016; Levasseur et al. 2016)) support a use of alternative metric values taking climate-carbon cycle feedbacks consistently into account (i.e. Table 8.SM.15 in the Supplementary Material of IPCC AR5), even though alternative values are available only for a subset of the components of interest. Now, the paper reveals that the approach to incorporating climate-carbon cycle feedbacks for non-$CO_2$ components adopted in IPCC AR5 was actually wrong because the natural carbon sinks are assumed inactive for the additional $CO_2$ release through climate-carbon cycle feedbacks (e.g. Figure 2). This finding essentially disqualifies all the alternative metric values in IPCC AR5.

Given the situation above in the recent past, this paper may create a new confusion among metric users dealing with climate and environmental policies and assessments. I would therefore request a more detailed clarification of what has happened and what should be done for the metric values in IPCC AR5 in their view. I think that this paper is a right place to do so because some of the authors have been closely involved in the writing of the metric section of IPCC AR5.

Hopefully this comment can be taken in a constructive way, but the paper can be more explicit about why the treatment of climate-carbon cycle feedbacks ended up with being inconsistent in IPCC AR5. The paper describes how it is inconsistent in sufficient details (e.g. Page 5, Lines 3-9), but it is unclear to me why this has happened. For

instance, why was it not possible to estimate an IRF for CO2 response without climate-carbon cycle feedbacks? If this were available, this might have allowed one to estimate metrics 'consistently' without climate-carbon cycle feedbacks. This might have been an alternative solution, if not a best one, in light of the inherent linear limitation in the IRF approach that is discussed in Section 5.2. In practice, it is probably not feasible to re-do an experiment requiring many models. But, looking back, was there a lack of coordination at the beginning? Furthermore, what about the method to account for climate-carbon cycle feedbacks for non-CO2 components in AR5? This method has not been sufficiently tested before the adoption and is based just on a section of one peer-reviewed paper (Collins et al. 2013), whose main contributions lie elsewhere. How was the ad-hoc decision process leading to the adoption of this approach made? Is there anything useful that can be learned from for the next IPCC report? What are the recommendations for metric values? I noticed that the paper does contain some text recommending the new estimates (page 12, lines 15-17), but it is buried in the middle of the paper and I am not sure what are the intentions. I raised some of the questions that may arise if the paper is officially published, although not all of them may not have to be answered in this paper. Clarifications suggested here should be helpful for metric applications, and ultimately the IPCC AR6.

The reviewer raises a fascinating and important point, but we fear, well beyond the scope of this paper. Essentially, the reviewer is passing comment on IPCC processes. The IPCC assesses the literature, and by doing so, places appropriate confidence on different findings. This comment is far broader than just the feedback value from Collins et al. (2013). The GWPs have been update in all five ARs, sometimes due to shifting background concentrations and sometimes due to improved scientific understanding. Thus, users of metrics should have an expectation that GWPs (and GTPs) will change in the future, due to both shifting background concentrations and improved scientific understanding. The inconsistency with the feedbacks has occurred in all ARs, and neither the literature (including reviewers) nor the IPCC has elevated this issue sufficiently for new analysis to be performed, until Collins et al. (2013). Yes, one could question
the IPCC and scientific community processes, a worthy endeavor but well beyond the scope of this paper. But also, the IPCC AR5 WG1 Chapter 8 has clearly noted that values change and will continue to change.

As to making recommendations, following the same logic that our paper should remain a research paper, we have decided not to make any. We now believe that our initial recommendation, although motivated by being the most up-to-date and comprehensive in terms of modeling, may not be the best – as shown by the comment by M. Sarofim and colleagues. Now we limit our paper to a short discussion as to why one would want to exclude or include the feedback, considering this is not our choice to make. We understand this won't please the metric-user community, but it appears that the choice is ultimately more a political choice than a scientific one (just as for the time horizon).

Relevant text added: "Therefore, the choice of including or excluding the feedback ultimately depends on the user's needs. On the one hand, for the sake of simplicity and transparency, the feedback could be excluded from the evaluation of GWPs, since it avoids the trouble of the five convolutions shown in figure 4. On the other hand, if absolute (e.g. time-varying) metrics are used as a first-order model of climate change, one may prefer including the climate-carbon feedback to have a better representation of the system."

Page 1, Lines 11-14 The argument concerns only a set of works using IRF to estimate emission metrics. There are also a body of relevant works based on other approaches like simple climate models (e.g. (Tanaka et al. 2009)) and more complicated ones (e.g. (Gillett and Matthews 2010)). These particular studies do consider climate-carbon cycle feedbacks to calculate emission metrics. The statement can be revised to be more restrictive.

We have added the following sentence to the first paragraph of the 'mathematical

framework' section (in which IRFs are presented as a means to calculate metrics): "Note that emission metrics can also be estimated thanks to complex model simulations (e.g. Tanaka et al., 2009; Sterner and Johansson, 2017), with the strong caveat that the approach lacks the simplicity and transparency of the IRFs."

We also note that part of the discussion (section 5.2) is dedicated to the interest of model-based metric estimates (that can include feedbacks in a much easier way than IRFs).

Page 2, Line 2 Another area that I could think of is the ecosystem community (e.g. (Neubauer and Megonigal 2015)).

Yes. Added.

Page 2, Line 13 If there is any reference to support this statement for the last century, please add.

It was unclear that the references to support this statement were the same as for the next sentence. So we have slightly altered the two sentences to put the references at the right place.

Page 2, Line 21 "whose" instead of "which"?

Changed.

Page 3, Line 9 I don't think the underlying models exhibit a hysteresis within the range of IRF calibrations.

Yes they do! The inertia of the simplest IRF (one decaying exponential) is enough to exhibit hysteresis. If a symmetric forcing is applied to any of the IRFs used in this

paper, the resulting response will show hysteresis if looked at in the (forcing, response) plane.

Page 4, Line 6 In practice, this pulse emission is large. As in Appendix A, it is 100 GtC in the case of CO2.

Agreed. A sentence has been added: "Note that the assumption of a very small pulse may be inconsistent with the way the IRFs are actually derived, as it is currently the case for CO2 (see appendix A)."

Page 5, Line 26 Should it be a(t') instead of a(t) in the integral?

Yes. Corrected.

Page 7, Line 16 It would be helpful if the authors provide a few sentences on how climate-carbon cycle feedbacks are modeled in OSCAR, rather than just a reference. Do the feedbacks act only on soil carbon? What about NPP? Do they directly affect the ocean carbon uptake?

Done:

"OSCAR includes the following climate-carbon feedbacks: the effect of temperature and precipitation change on net primary productivity of land ecosystems, their heterotrophic respiration, and the rate of occurrence of wildfires; and the effect of temperature change on the carbonate chemistry and the stratification of the surface ocean."

Page 10, Line 3 Please elaborate on how this equation was derived.

The way we derive this equation is explained in the three previous paragraphs. We find difficult to elaborate further. But we have extended one sentence in the third paragraph

to make clearer where the Dirac-$\delta$ function comes from, and we have put the final equation in a separated paragraph starting with "based on the above" to improve clarity.

Page 10, Lines 9-10 There are many earth system processes that are nonlinear. As something that has been discussed intensively before, I would point out the buffering of ocean CO2 uptake under rising atmospheric CO2 concentration. But this nonlinearity can be modeled by a revised IRF approach that treats the atmosphere and the mixed layer as one box (Hooss et al. 2001).

True, but our point was about reversibility. We have added some clarification: "This is however likely unrealistic, given all the existing processes, such as vegetation migration (e.g. Jones et al., 2009) or permafrost thawing (e.g. Koven et al., 2011), that can produce some degree of irreversibility in the system but are ignored here."

Page 11, Lines 4-5 Related to the comment above, the authors should refer to the relevant debate on the linear limitation of IRF (Joos et al. 1996; Hooss et al. 2001). A detailed biogeochemical discussion is given in Section 2.1.2 of (Tanaka et al. 2007).

We disagree that this section is the place where to mention this 'debate'. We demonstrate that the IRF we derived is only an approximation and therefore has a limited domain of validity, and later (in the conclusion) we remind the reader that more complex models should be used in some cases. We find this sufficient as our paper is not a review on IRFs. Note also that we do discuss the interest of more complex model in sections 5.2.

Page 12, Line 2 This should be "figures 5 and 6" because there is no figure 7 in the current manuscript.

Corrected.

[Figure]

Page 12, Lines 15-17 Related to my major comment, if this is really a recommendation for metric users, this needs to be more highlighted in the text. Metric users would otherwise be left wonder what are the values that should be used for applications.

As explained above, we have decided to not recommend any particular metric. This is not the goal of this paper.

Page 13, Line 20 This is just a minor note, but TOTEM (Ver et al. 1999; Mackenzie et al. 2011), which is one of the models used to derive the IPCC AR5 IRF (Joos et al. 2013), accounts for nitrogen and phosphorus limitations.

Yes. Although this does not play any role in the experimental setup of Joos et al. (2013) – or in ours – since there is no N or P deposition during the establishment of the IRF; just as there is no land-use change. And so we argue that these three drivers (and others) are therefore not accounted for in the IRF, even if the response has been calibrated on a model that in principle includes the drivers.

Page 14, Lines 29-30 I fully agree with this statement.

Thank you.

Page 15, Lines 16-17 I am coming back to the first minor comment. Although I some-what hesitate to repeat this point because of the conflict of interest, the paper should discuss studies that estimate emission metrics based on models other than IRFs at least at some length. Examples are (Manne and Richels 2001; Tanaka et al. 2009; Gillett and Matthews 2010; Reisinger et al. 2010; Johansson 2012; Smith et al. 2012; Tanaka et al. 2013; Sterner et al. 2014), and there are many more. The current manuscript narrowly focuses on IRF-based studies. I believe that adding more relevant studies should enrich the discussion in this paper and make the argument more

convincing.

We have added a reference to the very recent and only paper we know of that is a model-based study of the climate-carbon feedback in emission metrics (Sterner and Johansson, 2017).

Page 27 Figure 4 is not discussed in the paper

It is now. It was just a numbering issue.

---

## Author Comment (AC5) · 2 Mar 2017

In addition to the changes made to respond to the referee and short comments, we made two other changes to the paper.

First, we now refer to a very recent paper by Sterner and Johansson (2017) which is a model-based investigation of the impact of the climate-carbon feedback on emission metrics. Their conclusions are qualitatively the same as ours.

Second, with the aim of proposing the most up-to-date metrics, in addition to the update of the climate IRF, we now also include an update of the radiative efficiencies of $CO_2$, $CH_4$ and $N_2O$ (Etminan et al., 2016). Therefore, we have added a new row in table 2 and some new text:

[Figure]

"In table 2 (fifth row), we provide another set of relative metrics, similar to the previous one in that it includes the feedback response calibrated on OSCAR and the updated climate IRF, but it also includes an update of the radiative efficiencies of CO2, CH4 and N2O (Etminan et al., 2016). The new radiative efficiency of CO2 differs by +2%, that of CH4 by +14%, and that of N2O by -3%. These changes logically impact the GWPs and the GTPs, since both metrics are function of the $\varphi^x$ parameters. The change is substantial for CH4: in most cases more so than the update of the climate IRF. Notably, the update of the radiative efficiency of CO2 – being the reference gas in relative metrics – implies a change in the metrics' values of all species, even those whose own radiative efficiency are not changed. These results show that the first-order processes (here, the radiative forcing) may have more impact on the metrics than second-order processes such as the climate-carbon feedback."

The proposed revised manuscript – with track changes – is in attachment.

Please also note the supplement to this comment:
http://www.earth-syst-dynam-discuss.net/esd-2016-55/esd-2016-55-AC5-supplement.pdf
* * *